

# Representation of the Community Earth System Model (CESM1) CAM4-chem within the Chemistry-Climate Model Initiative (CCMI)

S. Tilmes[1], J.-F. Lamarque[1], L. K. Emmons[1], D. E. Kinnison[1], D. Marsh[1], R. R. Garcia[1], A. K. Smith[1], R. R. Neely[1,2], A. Conley[1], F. Vitt[1], M. Val Martin[3], H. Tanimoto[4], I. Simpson[5], D. R. Blake[5], and N. Blake[5]

[1]National Center for Atmospheric Research, Boulder, Colorado, USA
[2]National Centre for Atmospheric Science and the Institute of Climate and Atmospheric Science, University of Leeds, Leeds, UK
[3]The University of Sheffield, Sheffield, S1 3JD, UK
[4]National Institute for Environmental Studies, Tsukuba, Ibaraki 305-8506, Japan
[5]University of California, Irvine, CA, USA

*Correspondence to:* S. Tilmes (tilmes@ucar.edu)

**Abstract.** The Community Earth System Model, CESM1 CAM4-chem has been used to perform the Chemistry Climate Model Initiative (CCMI) reference and sensitivity simulations. In this model, the Community Atmospheric Model Version 4 (CAM4) is fully coupled to tropospheric and stratospheric chemistry. Details and specifics of each configuration, including new developments and improvements are described. CESM1 CAM4-chem is a low top model that reaches up to approximately

40 km and uses a horizontal resolution of 1.9 ° latitude and 2.5 ° longitude. For the specified dynamics experiments, the model is nudged to Modern-Era Retrospective Analysis For Research And Applications (MERRA) reanalysis. We summarize the performance of the three reference simulations suggested by CCMI, with a focus on the observed period. Comparisons with elected datasets are employed to demonstrate the general performance of the model. We highlight new datasets that are suited for multi-model evaluation studies. Most important improvements of the model are the treatment of stratospheric

aerosols and the corresponding adjustments for radiation and optics, the updated chemistry scheme including improved polar chemistry and stratospheric dynamics, and improved dry deposition rates. These updates lead to a very good representation of tropospheric ozone within 20% of values from available observations for most regions. In particular, the trend and magnitude of surface ozone has been much improved compared to earlier versions of the model. Furthermore, stratospheric column ozone of the Southern Hemisphere in winter and spring is reasonably well represented. All experiments still underestimate CO

most significantly in Northern Hemisphere spring and show a significant underestimation of hydrocarbons based on surface observations.





## 1 Introduction

The Chemistry Climate Model Initiative (CCMI) coordinates evaluation and modeling activieties for both tropospheric and stratospheric global chemistry-climate models. The CCMI-1 model experiments include three reference and several sensitivity experiments to evaluate the performance of chemistry-climate models in the troposphere and stratosphere for past and present

conditions (REFC1 and REFC1SD), and to identify future climate trends (REFC2) (Eyring et al., 2013). The REFC1SD simulation differs from the REFC1 simulation in that the dynamics are specified from reanalysis. Comprehensive tropospheric and stratospheric chemistry has been integrated into the Community Atmospheric Model Version 4 (CAM4-chem) of the Community Earth System Model (CESM1) and shows a reasonable representation of present-day atmospheric composition in the troposphere (Lamarque et al., 2012; Tilmes et al., 2015) and stratosphere (Lamarque et al., 2010). This model is therefore well

suited to participate in the CCMI model intercomparison project.

The purpose of this paper is to summarize the model configurations that were used to perform the reference CCMI model experiments (Section 2) including physics, dynamics, the chemical mechanism and aerosol description, as well as a summary of newly integrated diagnostics. We also describe issues that have been identified after the simulations were performed and their likely impacts. In addition, we summarize the global performance of the model in Section 3, and evaluate selected diagnostics

based on observational data sets in Section 4. We employ existing and new datasets to evaluate the general performance of the model. Improvements in comparison to earlier versions of the model are discussed in the Conclusions.

## 2 Model description

CESM is a fully coupled Earth System model, which includes atmosphere, land, ocean, and sea-ice components. All CCMI simulations are carried out with the same model code that is based on CESM version 1.1.1 (CESM1) (Neale et al., 2013), with

modifications discussed below. The configuration of the model used here fully couples the Community Atmosphere Model Version 4 (CAM4), the Community Land model Version 4.0 (CLM4.0), the Parallel Ocean Program Version 2 (POP2), and the Los Alamos sea ice model (CICE Version 4). The land model does not include an interactive carbon or nitrogen cycle and only the atmospheric and land components are coupled to the chemistry. The climatological present-day land cover is used for all simulations.

**2.1 The atmosphere model**

Detailed information about the physics of the atmosphere model used here are described in Neale et al. (2013) and Richter and Rasch (2008), and also summarized in Lamarque et al. (2012, and references therein). In summary, deep convection is treated by the Zhang and McFarlanle (1995) scheme, with improvements as described in Richter and Rasch (2008) and Neale et al. (2008). The photolysis calculation uses a look-up table between 200 and 750 nm and online calculations for wavelengths

$< 200$ nm. Attenuation of the spectral irradiance above the model top is calculated using the approach of Kinnison et al. (2007); Lamarque et al. (2012).



Processes in the planetary boundary layer are represented using the Holtslag and Boville (1993) parameterization. Wet deposition of gas and aerosol compounds is based on Neu and Prather (2012), as described in Lamarque et al. (2012). In this version of CAM4-chem all aerosols in the cloudy fraction of the grid cell are assumed to reside within cloud droplets and are removed in proportion to the cloud water removal rate. Aerosols directly impact the radiation and chemistry, but do not change

the radiative properties of clouds (i.e., no representation of the aerosol indirect effects is included).

### 2.1.1 Model grid

For all CCMI reference simulations, CESM1 CAM4-chem uses a horizontal grid with a resolution of 1.9 ° x 2.5 ° (latitude by longitude), and uses the finite volume dynamical core. The top of the model is located at 3 hPa (about 40 km). The vertical coordinate is sigma (terrain-following) in the troposphere, switching over to isobaric above 100 hPa; the vertical resolution of the

10 model depends on the configuration of the experiment. The atmosphere model, CAM4, makes use of 2 different configurations, the free running (FR, with 26 vertical levels) and the specified dynamics (SD, with 56 vertical levels adopted from the analysis fields), see Lamarque et al. (2012). For the SD configuration, internally derived meteorological fields are nudged every time step of 30 minutes by 1% towards reanalysis fields (equivalent to a 50 h Newtonian relaxation time scale for nudging) from Modern-Era Retrospective Analysis For Research And Applications (MERRA) reanalysis (http://gmao.gsfc.nasa.gov/merra/)

(Rienecker et al., 2011).

### 2.1.2 Quasi-Biennial Oscillation

The SD configuration of the model incorporates the observed Quasi-Biennial Oscillation (QBO), which is present in the meteorological analysis fields. The limited vertical resolution of the FR model configurations does not allow for the generation of an internal QBO in CAM4-chem. Therefore, for the FR CCMI experiments, REFC1 and REFC2, the QBO is imposed in

the model by relaxing equatorial zonal winds between 86 hPa and the model top to the observed interannual variability, following the approach by Matthes et al. (2010). Here, we vary the QBO phase between eastward and westward phase using an approximate 28-month period, similar to what was done by Marsh et al. (2013).

### 2.1.3 Improved gravity wave representation

The representation of sub-gridscale gravity waves (GW) in CAM was formerly limited to orographic gravity waves using the

25 parameterization adapted from McFarlanle (1987). In the present simulations, the parameterizations of non-orographic gravity waves generated by convection (Beres et al., 2005) and fronts (Richter et al., 2010), which were developed for the Whole Atmosphere Community Climate Model (WACCM), are also included.

In addition, we have added another gravity wave module to represent the waves with large horizontal wavelengths that are often observed in the stratosphere (e.g., Zink and Vincent, 2001). The new GW module is adopted from the inertia-gravity wave

(IGW) parameterization developed by Xue et al. (2012) for an interactive QBO. The formulation includes the impact of the Coriolis force on gravity wave propagation and breaking. Rather than applying it in the equatorial region, as done by Xue et al.



(2012), we use a more general mechanism for determining sources; gravity waves are triggered by the same frontal threshold used for the mesoscale gravity waves (Richter et al., 2010). This has the impact of shifting the bulk of the waves from the tropics to middle and high latitudes. In the current implementation, gravity waves have a narrow phase speed spectrum (-20 to 20 m/s) and long horizontal wavelength (1000 km). The momentum forcing associated with this module particularly impacts

the winter stratosphere. In the Southern Hemisphere (SH), it enhances downwelling and increases the winter stratospheric temperature, which in previous simulations was substantially colder than observed.

However, it was found, that the version of the IGW parameterization used for the performed experiments has a narrow IGW spectrum centered on zero phase velocity instead of being centered on the speed of the background wind at the GW launch level, as in the standard GW parameterization. Even with this shortcoming, the model produces a much improved temperature

evolution in the stratosphere, in particular in the SH high latitudes compared to earlier versions. This results in a well resolved ozone hole in winter and spring over Antarctica. No significant changes are expected from a corrected IGW parameterization for the troposphere.

### 2.1.4 Tropospheric aerosols

CAM4-chem runs with the bulk aerosol model (BAM), which simulates the distribution of externally-mixed sulfate, black

carbon (BC), primary organic carbon (OC), sea-salt and dust, as described in Lamarque et al. (2012). The dust emissions are calibrated so that the global dust aerosol optical depth (AOD) is about 0.025 to 0.030 (Mahowald et al., 2006). The distribution of sea-salt and dust are described using four size bins. In CAM4-chem, the formation of secondary organic aerosols (SOA) is coupled to the chemistry and biogenic emissions. SOA are derived using the 2-product model approach using laboratory determined yields for SOA formation from monoterpene, isoprene and aromatic photooxidation, as described in Heald et al.

(2008). The aging process of BC and OC from hydrophobic to hydrophilic is included through a specified conversion timescale. For all aerosol species, the size distributions are specified as in Lamarque et al. (2012). Aerosols interact with the gas-phase chemistry through heterogeneous reactions that depend on the available surface area density (SAD), as discussed below. For the tropospheric SAD calculation, sulfate, hydrophilic black carbon and primary organic carbon, nitrates are included, where SOA has not been included. This may lead to a significant underestimation of tropospheric SAD in the experiments.

### 2.1.5 Representation of aerosols in the stratosphere

Aerosol mass, heating rates and SAD are revised in this version compared to earlier configurations. Most significantly, the model uses a new stratospheric aerosol and SAD dataset, derived based on observations, to force models participating in CCMI (Eyring et al., 2013). In addition, in order to fully utilize the aerosol size information provided by the new model input file, the optics in the radiative transfer code associated with CAM4 (i.e., CAMRT) (Neale and al., 2010) have been modified

to include a lookup table for aerosol effective radius in the shortwave radiation scheme. The new description leads to an updated representation of volcanic heating for REFC1 and REFC2, whereas in REFC1SD volcanic heating is included through the nudged temperature fields. See Neely et al. (2015, in prep.) for a full description of changes to the stratospheric aerosol scheme.



### 2.1.6 Coupling to the land model

Dry deposition velocity for tracers in the atmosphere are calculated online in CLM. An updated calculation is used, where leaf and stomatal resistances are coupled to the leaf area index (LAI) and are also linked to the photosynthesis provided by the land model, as described by Val Martin et al. (2014).

5  Biogenic emissions are calculated online in CLM using the Model of Emissions of Gases and Aerosols from Nature (MEGAN), version 2.1 (Guenther et al., 2012). The implementation of MEGAN in this version differs from the description of Guenther et al. (2012) by using the LAI from the previous model timestep instead of the average of the previous 10 days, and by using a fixed $CO_2$ mixing ratio, instead of the simulated atmospheric value, in the calculation of the $CO_2$ inhibition effect on isoprene emissions.

### 10  2.2 Chemical mechanism and aerosol description

The chemical mechanism of CAM4-chem includes 169 species, listed in Table A1. Depending on the chemical lifetime of each species, an explict or semi-implicit solver is used. Different species experience wet and/or dry deposition, as also listed in Table A1. Furthermore, 14 artificial tracers are implemented as recommended by CCMI (Eyring et al., 2013): $NH_5$, $NH_{50}$, $NH_{50W}$, $AOA_{NH}$, $ST80_{25}$, $CO_{25}$, $CO_{50}$, $SO_{2t}$, SF6em, $O_3S$, E90, $E90_{NH}$, $E90_{SH}$. $O_3S$ is a stratospheric ozone tracer that represents

the amount of ozone in the troposphere with its source in the stratosphere. $O_3S$ is set to stratospheric values at the tropopause, and experiences the same loss rates as ozone in the troposphere, as defined by CCMI. Following the CCMI recommendation, dry deposition is not included, which will lead to an overestimation of $O_3S$ in the lower boundary layer when compared to ozone (which is dry deposited).

The chemical mechanism, is based on the Model for Ozone and Related chemical Tracers (MOZART), version 4 mechanism

for the troposphere (Emmons et al., 2010). It further includes extended stratospheric chemistry (Kinnison et al., 2007) and updates, as described in Lamarque et al. (2012) and Tilmes et al. (2015). The reactions include photolysis, gas-phase chemistry, and heterogeneous chemistry, in both troposphere and stratosphere. Furthermore, tropospheric aerosols that enter the stratosphere are promptly removed, since the aerosol burden in the stratosophere is prescribed. The complete chemical mechanism is listed in Table A2 and incorporates all the latest updates.

Reaction rates are updated following JPL2010 recommendations (Sander et al., 2011). Bromoform ($CHBr_3$) and dibromomethane ($CH_2Br_2$) were added to the model to represent the stratospheric bromine loading from very short lived (VSL) species. The surface volume mixing ratio for these two VSL species was set globally to 1.2 ppt (i.e., 6 ppt total bromine). This approach adds an additional $\approx$ 5 ppt of inorganic bromine to the stratosphere. The resulting stratospheric total inorganic bromine abundance (for present day conditions) from both long-lived and VSL species is $\approx$ 21.5 ppt. Besides the current

Lower Boundary Condition (LBC) approach for VSL species, CAM4-Chem can be also configured with a Full-VSL mechanism, including detailed gas-phase halogen chemistry mechanism, geographically and time-dependent distributed sources of 9 halocarbons and improved representation of heterogeneous recycling and removal rates in the troposphere (Fernandez et al., 2014; Saiz-Lopez et al., 2014).



Details on updated reactions and processes for chemistry in the polar stratosphere are described in Wegner et al. (2013) and Solomon et al. (2015).

Lightning $NO_x$ is parameterized following Price and Vaughan (1993); Price et al. (1997). The global amount of produced lightning $NO_x$ is scaled differently for the SD and the FR experiments due to differences in the meteorology to ensure values of approximately 3-5 Tg N/yr for present day conditions.

Diagnostics of the tropospheric ozone production and loss rates are explicitly calculated in adding the listed reaction rates r of two species A and B, r(A-B), as well as the photolysis reaction of ONITR (defined as lumped organic nitrate species that includes nitrates derived from the OH- and $NO_3$-initiated oxidation of isoprene and terpenes, and related species), called jonitr:

$$
\begin{aligned}
O3 - Prod = &\, r(NO - HO2) + r(CH3O2 - NO) + r(PO2 - NO) + r(CH3CO3 - NO) + \\
&\, r(C2H5O2 - NO) + .92 * r(ISOPO2 - NO) + r(MACRO2 - NOa) + r(MCO3 - NO) + \\
&\, r(C3H7O2 - NO) + r(RO2 - NO) + r(XO2 - NO) + .9 * r(TOLO2 - NO) + \\
&\, r(TERPO2 - NO) + .9 * r(ALKO2 - NO) + r(ENEO2 - NO) + r(EO2 - NO) + \\
&\, r(MEKO2 - NO) + 0.4 * r(ONITR - OH) + jonitr
\end{aligned}
$$

$$
\begin{aligned}
O3 - Loss = &\, r(O1D - H2O) + r(OH - O3) + r(HO2 - O3) + r(C3H6 - O3) + \\
&\, .9 * r(ISOP - O3) + r(C2H4 - O3) + .8 * r(MVK - O3) + 0.8 * r(MACR - O3) + \\
&\, r(C10H16 - O3)
\end{aligned}
$$

These are defined based on the rate-limiting terms for the gas phase reactions of the $O_x$ family ($O_3$, O, O1D, $NO_2$), not including $O_2 + h\nu \rightarrow 2O$ production, $O_x$, $ClO_x$, and $BrO_x$ losses, and are therefore not valid for the stratosphere. The sum of those rates are very similar to the explicit calculation of the net chemical change of ozone (as listed in Table A2).

### 2.3 Experimental Setup

The reference experiments are set up according to the CCMI recommendation, including surface and altitude dependent emissions, and lower boundary conditions. The three reference experiments are performed with the recommended emissions; for REFC1 and REFC1SD. Anthropogenic and biomass burning emissions are from the MACCity emission data set (Granier et al., 2011), while for REFC2, emissions are taken from AR5 (Eyring et al., 2013) (see Figure A1). Biogenic emissions are calculated by MEGAN, as described in Section 2.1.6.

The REFC1SD experiment is nudged to analyzed air temperatures, winds, surface fluxes, and surface pressure, and uses the Hadley Centre Global Sea Ice and Sea Surface Temperature (HadISST) observed time-dependent data set for sea surface temperatures (SSTs) and sea ice. The REFC1 experiment also uses prescribed SSTs and sea ice, while the REFC2 simulation calculates temperatures in the ocean and atmosphere. We have carried out one simulation for REFC1SD, and an ensemble of three members for each REFC1 and REFC2.





The solar cycle is prescribed using observed daily fields for the years until 2010. For the future period in REFC2, we follow the CCMI recommendation and repeat a sequence of the last four solar cycles (20-23), as defined in http://solarisheppa.geomar.de/ccmi.

### 2.3.1 Initial conditions and spin-up

CAM4-chem initial conditions for the three REFC1 and REFC2 ensemble members are taken from 3 realizations of CESM1-
5 WACCM 20th Century ensemble for CMIP5 (Marsh et al., 2013). The spin-up period started in 1950 and ran until 1959. The experiments simulated the years 1960 to 2010 (REFC1) and 1960 to 2100 (REFC2). Initial conditions for the REFC1SD simulation are taken from the first REFC1 ensemble member in 1975. The spin-up of this experiment covered the years 1975 to 1979, repeating 1979 meteorological analysis for each year. The experiment was performed between 1980 and 2010.

### 2.3.2 Lower boundary conditions

For all of the three reference experiments the same monthly and annually varying lower boundary conditions are used based on the Representation Concentration Pathway 6.0 (RCP6.0) Coupled Model Intercomparison Project Phase 5 (CMIP5) future projection (Taylor et al., 2012). We prescribe $CO_2$, $N_2O$, $CH_4$, as well as the following halogen species based on the CCMI recommendations: $CCl_4$, $CF_2ClBr$, $CF_3Br$, CFC11, CFC113, CFC12, $CH_3Br$, $CH_3CCl_3$, $CH_3Cl$, $H_2$, HCFC22, CFC114, CFC115, HCFC141b, HCFC142b, $CH_2Br_2$, $CHBr_3$, H1202, H2402, $SF_6$. A North-South gradient was added for $CH_3Br$,
HCFC22, HCFC141b, HCFC142b, based on the HIAPER (High-Performance Instrumented Airborne Platform for Environmental Research) Pole-to-Pole Observations (HIPPO) (Wofsy et al., 2011), (Mijeong Park, pers. comm.).

## 3 Model performance

### 3.1 Global diagnostics

The general state of the model is investigated by comparing diagnostics of globally averaged values between different model
experiments that are averaged between 1995 and 2010 (Table 1). The global surface temperatures (TS) of all three experiments are in agreement within 0.15 K for the observed period (Table 1). REFC1SD land temperature (TS land) is on average 0.25 K higher than for REFC1 and 0.15 K higher than for the REFC2 experiments (Table 1). The largest deviations occur over high latitudes (not shown). In the REFC1SD experiment, low cloud fraction is significantly larger than in the other experiments, which results in a much smaller shortwave cloud forcing (SWCF).

Differences in clouds and land surface temperatures between the reference experiments result in different biogenic emissions of volatile organic components (VOCs) (Figure 1). REFC1SD biogenic emissions are about 10% lower than derived in the REFC1 experiment and about 15% lower than in the REFC2 experiment. The emissions differ the most in summer during their peak (Figure 1, bottom row). Other differences in the REFC1 and REFC2 VOC emissions arise from different anthropogenic and biomass burning emissions, while biogenic emissions differ by less than 10% (Table 1). Despite the variation in the
reference experiments, biogenic emissions are in agreement with earlier estimates (e.g., Young et al., 2012).



The performance of tropospheric chemical variables (Table 1) is similar to earlier studies (e.g., Tilmes et al., 2015). Methane lifetime is low compared to observational estimates of 11.3 years (Prather et al., 2012). Ozone budgets, including ozone burden, stratosphere troposphere exchange, and budgets of carbon monoxide (CO), are in agreement with earlier model studies (Young et al., 2012). Aerosol burdens of primary organic matter (POM) and secondary organic aerosols (SOA) are low, but

within the spread of other model results (Tsigaridis et al., 2014). The $SO_4$ burden with 0.45 to 0.51 TgS and the lifetime of 3.0 to 3.5 days is somewhat low compared to the Aerocom multi-model mean of 0.66 TgS and 4.12 days, respectively (e.g., Liu et al., 2012). The dust optical depth is with around 0.04 somewhat higher than suggested by Mahowald et al. (2006).

### 3.2 Trends of tropospheric components

Time varying emissions of ozone precursors and aerosols impact the oxidation capacity of the atmosphere. In the following, we

discuss the evolution of different chemical species and surface area density in the tropical troposphere between 30° S–30° N, tropospheric methane lifetime, and stratospheric column ozone (Figure 2), since methane is mostly controlled by processes in the Tropics. Increasing nitrogen (N), CO and VOC burdens between 1960 and 1990 result in increasing tropospheric ozone with the strongest trend between 1960 and 1990. Increasing aerosols between 1960 and 1990 result in an increase in SAD, with little change after 1990. Together with the increase in CO burden, this results in a decrease of OH. On the other hand, decreasing

stratospheric column ozone between 1960 and 2010, and increasing column ozone combined with the increasing nitrogen oxides ($NO_x$) burden and methane emissions, increases tropospheric OH (e.g., Murray et al., 2014). Both counteracting effects on OH result in little change in methane lifetime between 1960 and 1990. After 1990, SAD, as well as CO and VOC trends are leveling off, but nitrogen and ozone burdens are still increasing, partly due to increasing lightning $NO_x$ production (not shown). This results in a decreasing trend in methane lifetime after 1990 for all reference experiments.

The burden of chemical tracers differ between REFC1SD and REFC1/ REFC2 (Figure 2). Variations in emissions and atmospheric dynamics, including surface temperature, clouds, and convection, influence the chemical composition of the atmosphere. Exchange processes between the upper troposphere and lower stratosphere are also different in the model experiments and impact ozone. For instance, larger ozone mixing ratios in the upper troposphere in the REFC1SD experiment results in a higher oxidation capacity of the troposphere and therefore a shorter lifetime of methane compared to the other experiments.

Besides a continuous decrease, the stratospheric ozone column shows a significant drop after major volcanic eruptions. This is expected due to an increase in stratospheric SAD after the eruption, which causes enhanced halogen activation, resulting in ozone depletion (see Figure 2).

### 4 Evaluation against selected diagnostics

The purpose of this section is to give an overview of selected variables and diagnostics that summarize the performance of

30 the model, including some of its shortcomings, in comparison to observations. Additional and more detailed investigations are expected in future multi-model comparison studies. We only discuss the performance of the reference experiments for past and present day. Model results from other sensitivity studies are not analyzed and will be discussed in future studies.





## 4.1 Ozone

Ozone is an important atmospheric tracer in both the troposphere and the stratosphere. In the troposphere and at the surface, ozone is an air pollutant and is impacted by various precursors, most importantly CO and $NO_x$. A reasonable performance of tropospheric ozone is required for air quality studies. In the stratosphere, ozone is strongly influenced by dynamics, photo-
chemistry, and catalytic reactions (e.g., WMO, 2010). The strength of the transport of stratospheric ozone into the troposphere follows a seasonal cycle controlled by the Brewer Dobson circulation (BDC). Shortcomings in the representation of the strength of the BDC and mixing processes between stratosphere and troposphere influence the performance of tropospheric ozone, as discussed below. In addition, ozone is an important greenhouse gas in the upper troposphere and lower stratosphere (UTLS) and influences tropospheric climate (e.g., WMO, 2014)).

### 4.1.1 Trends and seasonality of ozone

Ozone trends and seasonality in the reference experiments are compared to ozonesonde observations (Tilmes et al., 2012) in the free troposphere (at 500 hPa) and the boundary layer at 900 hPa. For Japan, we employ an additional climatology derived by Tanimoto et al. (2015), which is based on surface observations at five marine boundary layer sites from the Acid Deposition Monitoring Network in East Asia (EANET) for altitudes below 900 hPa, and a combination of the historical Measurements of
OZone and water vapour by in-service Aircraft (MOZAIC, URL: http://www.iagos.fr/mozaic) data (over Narita airport) and ozonesonde observations (at Tateno/Tsukuba) for altitudes between 472 and 616 hPa. We use an artificial stratospheric ozone tracer ($O_3S$) to identify differences in stratosphere troposphere exchange (STE) between different model experiments for four selected regions (see Figures 3 and 4).

In high northern latitudes, REFC1SD reproduces the magnitude and trend of ozone very well, including variability within
the standard deviation of the observations for all seasons, as shown in the example of the Northern Hemisphere (NH) Polar West region (Figure 3, first and second row). A very good agreement between the model experiment and ozonesondes also exists for Western Europe, with exception of the high bias between October and February at 500 hPa of 5-10 ppb (Figure 3, third and fourth row).

Results from REFC1 and REFC2 show larger deviations from the observations than REFC1SD over these two regions, which
are due in large part to differences in the amount of stratospheric ozone entering the troposphere for the different experiments (see Figure 3, right column, dashed lines). Discrepancies in ozone between the experiments can be explained by differences in $O_3S$ for the whole year at 500 hPa and for winter months at 900 hPa. During summer months, differences in chemical production at the surface for the different experiments seem to play an additional role and explain about 5-10 ppb of the deviations for Western Europe.

Selected ozonesondes over Eastern US and Japan are located further south and are more strongly influenced by tropical air masses and tropospheric intrusion in the lowermost stratosphere in particular in winter, as discussed in Tilmes et al. (2012). Each of the regions covers only two stations and so uses fewer observations for the different years than other regions, which increases the uncertainty of trends (Saunois et al., 2012).



Comparisons for Eastern US and Japan are illustrated in Figure 4. For Japan, we are using two datasets to compare to model results. Ozone mixing ratios and trends at 900 hPa over Japan using ozonesondes, as compiled by Tilmes et al. (2012), Figure 4 (black diamonds), largely differ from the climatology by Tanimoto et al. (2015), which is based on surface observations (black triangles). This is due to uncertainties in the ozonesonde observations at these altitudes, which should be treated with caution.

On the other hand, the two climatologies agree well in the free troposphere at 500 hPa.

For Eastern US and Japan the REFC1SD model experiment nicely reproduces the observed trend and magnitude of ozone within the variability of the observations at 900 hPa. The seasonal cycle for both regions are well reproduced. This significant improvement compared to earlier versions of the model is in part a result of the improved calculation of dry deposition rates, as discussed in Val Martin et al. (2014) over the U.S. . REFC1/REFC2 experiments show slightly larger values at 900 hPa in

comparison to the REFC1SD experiment particularly in winter, aligned with a larger $O_3$ contribution from the stratosphere, as determined by the $O_3S$ tracer (see Figure 4). At 500 hPa, ozone mixing ratios and trends are well reproduced for all experiments in summer. However, the model overestimates winter ozone mixing ratios in the upper troposphere.

### 4.1.2 Present-day ozone

A comparison with ozonesonde observations over different regions between 1995-2010 is presented in Figure 5. Besides some

15 differences in ozone compared to observations, as discussed above, all model experiments reproduce observed tropospheric ozone within about 20%. At 250 hPa, which is the UTLS at mid and high latitudes, REFC1SD overestimates ozone by up to 50%, particularly at mid latitudes in both hemispheres, while the other experiments show smaller deviations from the observations of about 20% or less. Tropical values at 50 hPa are overestimated by no more than 20% compared to observations for all the experiments, while ozone in the mid and high latitudes in the stratosphere agrees within 10% with observations.

Model results further agree well with HIPPO aircraft observations for profiles sampled from 85° N–65° S over the Pacific Ocean between 2009 and 2011 (Figure A2). In REFC1SD, lower troposphere values (1-2 km) are within the range of the observations, while for REFC1 and REFC2 ozone is overestimated by about 5 ppb in high northern latitudes, in particular in winter and spring, which points to a transport problem as discussed above. Some differences, especially at higher altitudes (7-8 km) are likely caused by the specific meteorological situation for the flight conditions compared to the climatological model

results.

The regional performance of tropospheric ozone in the model is further illustrated in Figure 6, comparing simulated ozone mixing ratios with ozone sondes and various aircraft observations at 3-7 km, as compiled in Tilmes et al. (2015). Besides the described differences between REFC1 and REFC1SD experiments, observed features, for example the summertime maximum of ozone over eastern Mediterranean/Middle East (Kalabokas et al., 2013; Zanis et al., 2014), are reproduced by the model.

The ozone gradient between mid latitudes and tropics is to the most part well captured, for example over Japan in summer. Regional differences in tropospheric ozone between the different model experiments have to be investigated in future studies.

We further perform comparisons of model results to a present day ozone climatology based on OMI and MLS satellite observations between 2004 and 2010, compiled by Ziemke et al. (2011), in the troposphere (Figure 7) and stratosphere (Figure 8). The model tropopause for this diagnostic is defined at the 150 ppb ozone level. The comparisons reveal additional character-



istics of the model performance compared to observations. Tropospheric column ozone is reproduced within $\pm$ 10 DU of the observations, with a close agreement to the satellite climatology within less than $\pm$ 5 DU in low and mid latitudes in spring and summer (Figure 7). All model experiments show a low bias in mid latitudes in the SH and high bias by 10-15 DU in the NH mid latitudes in winter and fall. NH tropospheric ozone is in general large in the REFC1 and REFC2 simulations compared to

5 the REFC1SD experiments, as discussed above.

Stratospheric ozone in all model experiments agree within $\pm$ 30 DU in mid and low latitudes compared to the satellite climatology (Figure 8). Larger deviations from the observations occur in the NH mid and high latitudes in winter and spring with a high bias of up to 60 DU. Ozone in the SH is within about 25 DU from the observations and is reasonably well reproduced by all model experiments, especially for the free running experiments.

## 4.2 Carbon Monoxide

Carbon monoxide, non-methane hydrocarbons and nitrogen dioxides are the most important precursors to the formation of tropospheric ozone. Carbon monoxide also impacts the oxidation capacity of the atmosphere and therefore methane lifetime. We compare the CO burden from different experiments to monthly and zonally averaged tropospheric column carbon monoxide derived from Measurements of Pollution in The Troposphere (MOPITT) Version 6 Level 3 satellite observations, as described

in Tilmes et al. (2015) (see Figure 9). The climatological averaging kernel and a priori is applied to both observations and model experiments in the same way.

The most obvious difference between observations and model results occurs in NH winter and spring. All model experiments are biased low by about a third relative to observations, similar to result from the Atmospheric Chemistry and Climate Model Intercomparison Project (ACCMIP) (Naik et al., 2013; Lamarque et al., 2012). In summer and fall, the CO representation

differs between different experiments, in agreement with differences in biogenic emissions. The lowest CO burden is simulated for the REFC1SD experiment, which also shows the lowest emissions of VOCs in summer (see Figure 1). This may translate into lower CO values in fall. Furthermore, the tropospheric OH burden is significantly larger in REFC1SD compared to the other experiments (not shown), which is consistent with more ozone in the tropical troposphere (see Figure 2).

Simulated CO column in the tropics agree with the satellite climatology within the interannual variability. However, the

25 model underestimate CO column in the SH for all the experiments, in particular in summer. In contrast, comparisons to HIPPO CO in-situ observations indicate very good agreement between CO mixing ratios in the SH over the remote region of the Pacific Ocean for most of the seasons (see Figure A3). Furthermore, CO mixing ratios are largely underestimated in March and April in comparison to the aircraft observations, consistent with the satellite comparison. Differences in CO will be investigated in more detail in future studies.

## 4.3 Hydrocarbons

Hydrocarbons are important tropospheric compounds that are emitted from vegetation, biomass burning and anthropogenic sources, including oil and gas extraction activities. They are important ozone precursors, influence the oxidation capacity of the atmosphere, and eventually form CO.



Ethane and other hydrocarbons have been measured using canister samples along coastal and island sites in the Pacific Ocean since 1984 typically every three months, December, March, June and September (Simpson et al., 2012); data are available at http://cdiac.ornl.gov/trends/otheratg/blake/blake.html. We have compiled a climatology using ethane mixing ratios between 1995 and 2010 that covers latitudes between 50° S and 75° N (shown in Figure 10). Comparisons to the three model experi-

ments reveal a very large underestimation of ethane mixing ratios by up to 5 times in spring. The smallest deviations occur in NH fall. These deviations are likely contributing to the underestimation of CO and overestimation of OH.

While there is significant uncertainty in the speciation of VOC emissions (e.g., Li et al., 2014), which could lead to this discrepancy, it is likely there is an underestimation of all VOC emissions. Globally, ethane concentrations have been declining since long-term global record-keeping began. Simpson et al. (2012) reported a 21% decline in global ethane concentrations

from 1984 to 2010, which is much smaller than the discrepancy between the model and observations.

### 4.4   Aerosols

A reasonable description of aerosols in climate models, including interactions with chemistry and clouds, is important for the representation of radiative processes. The aerosol optical depth, global aerosol burden of organic matter, black carbon, and sulfate aerosol, are global diagnostics to evaluate the general performance of aerosol processes (Table 1). This version of

CAM4-chem produces values for these diagnostics very similar to earlier model studies using CAM4-chem (e.g., Tilmes et al., 2015). Here, we focus on the evaluation of background black carbon in comparison to HIPPO observations. The HIPPO campaign between 2009 and 2011 provided a comprehensive data set of black carbon over the remote region over the Pacific. Black carbon results from the model are averaged over the same locations, and altitude levels and compared to the observations, as described above.

All model simulations show a very similar distribution (Figure 11), with only a few deviations from each other mostly in the SH. The model reproduces BC values in the SH and NH mid latitudes. A significant high bias in BC occurs in the tropics for all altitude levels and most seasons. Otherwise, the South-to-North gradient of BC is represented well, following the observed larger burden in the NH compared to the SH in March/April and June/July. The largest BC values in the NH spring are however underestimated. On the other hand, BC values in August/September, and partly November, are overestimated in the NH and in

March/April and June/July in the SH.

### 5   Conclusions

The CESM1 CAM4-chem model has been used to perform the CCMI reference and sensitivity simulations. This paper provides an overview of the model setup of the reference experiments, including a detailed description of new developments. The most important improvements of the model beyond what has been discussed in earlier studies (Lamarque et al., 2012; Tilmes et al.,

2015) are the treatment of stratospheric aerosols and the corresponding radiation and optics, which is important for the free running experiments (Neely et al., 2015). Further, the chemistry scheme has been updated to reaction rates of JPL 2010, and improved polar chemistry has been implemented (Wegner et al., 2013; Solomon et al., 2015). A new gravity wave description,



while implemented incorrectly in the code, led to an improved representation of the evolution of polar stratospheric ozone in the SH. The updated dry deposition scheme by Val Martin et al. (2014) resulted in a much improved ozone near the surface, as also shown in Tilmes et al. (2015), and leads to a very good representation of ozone mixing ratios and trends in the REFC1SD simulation.

Global model diagnostics are investiaged and a selected evaluation of key chemical species, including ozone, carbon monoxide, hydrocarbons, and black carbon is performed. We limit our evaluation to present day results of the REFC1SD, REFC1 and REFC2 experiments. Comparisons to observations are focused mostly on the troposphere. Nevertheless, stratospheric column ozone reproduces observed values, in particular in SH winter and spring, but overestimates values in the NH high latitudes.

    For the troposphere, near surface ozone mixing ratios and trends are very well reproduced and within 20% of the values
from ozonesonde and satellite observations throughout the troposphere. A high bias in mid and high northern latitudes for the REFC1 and REFC2 experiments can be explained by a stronger influence of stratospheric air masses compared to the REFC1SD simulation. This points to shortcomings in the stratosphere to troposphere exchange in the free running simulations. On the other hand, the specified dynamics model experiment shows an overestimation of ozone in mid latitude UTLS, as well as enhanced ozone in the upper tropical troposphere compared to the free running experiments. The impact of shortcomings in
the dynamical description of the model needs to be investigated in multi-model comparison studies.

    Some biases in the model have not been resolved compared to earlier versions of the model. CO is still biased low in all model experiments in the NH, especially in spring. Some differences between the experiments may be ascribed to differences in biogenic emissions. Correspondingly, methane lifetime is rather low compared to observational estimates, which is likely related to shortcomings in emissions, but also to too great oxidation capacity of the atmosphere. Significant shortcomings of
hydrocarbons (shown for ethane) are identified in particular in the NH. The North-to-South gradient of BC in the model is reproduced well in most seasons, while the fall and winter values in mid latitudes are often overestimated in mid latitudes. BC in the Tropics is largely overestimated for most seasons. This points to potential shortcomings in emissions, but also loss processes in the model.

## 6   Code and data availability

The model code of the documented simulations is based on the Community Earth System Model, CESM version 1.1.1 (CESM1), http://www.cesm.ucar.edu/models/cesm1.1/index.html. Modifications to the model code will be documented at http://www2.cesm.ucar.edu/models/scientifically-supported. The data of the simulations are available for download at the NCAR Earth System Grid (ESG) (https://www.earthsystemgrid.org/home.html ) and are submitted to the BADC database for the CCMI project.

*Acknowledgements.* We thank the HIPPO team for performing reliable aircraft observations used in this study. MERRA data used in this study have been provided by the Global Modeling and Assimilation Office (GMAO) at NASA Goddard Space Flight Center through the



NASA GES DISC online archive. The CESM project is supported by the National Science Foundation and the Office of Science (BER) of the US Department of Energy. The National Center for Atmospheric Research is funded by the National Science Foundation.



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



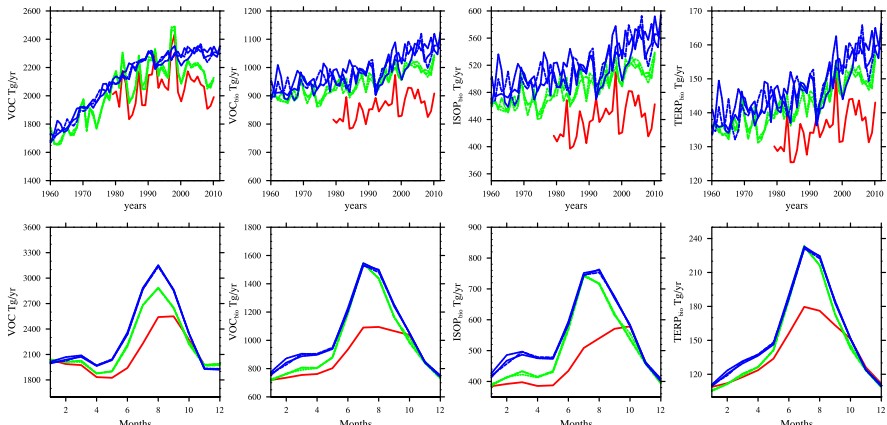

**Figure 1.** Global averaged surface emissions of total volatile organic compounds (VOCs) (first column), biogenic VOCs (second column), biogenic isoprene (third column), and biogenic terpenes (fourth column), for different experiments, REFC1SD (red), REFC1 (green), REFC2 (blue). The seasonal cycle of zonal averages between 1960 and 2010 are shown at the bottom row.

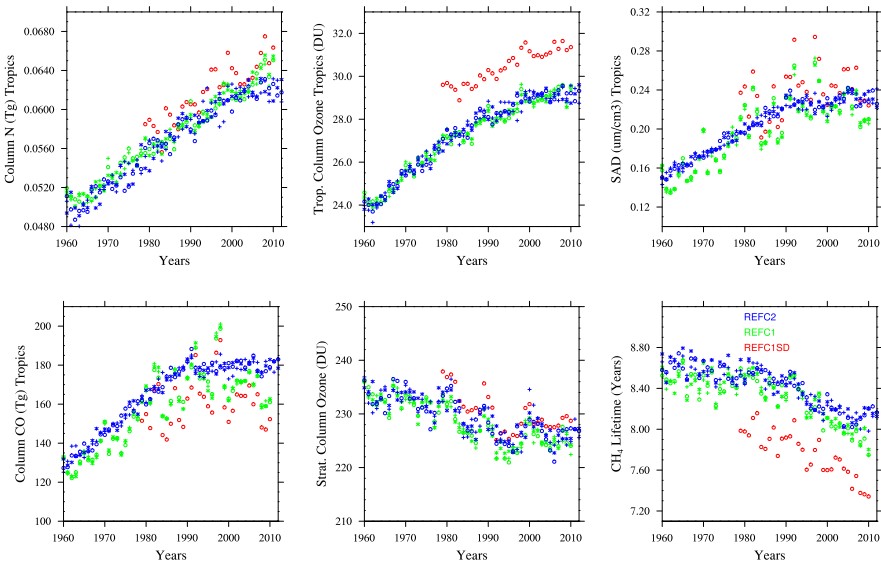

**Figure 2.** Timeseries of annually averaged column integrated tropospheric and tropical nitrogen, tropospheric ozone burden, and CO, in (30° S–30° N), tropical average of tropospheric surface area density, global stratospheric column ozone, and tropospheric methane lifetime.



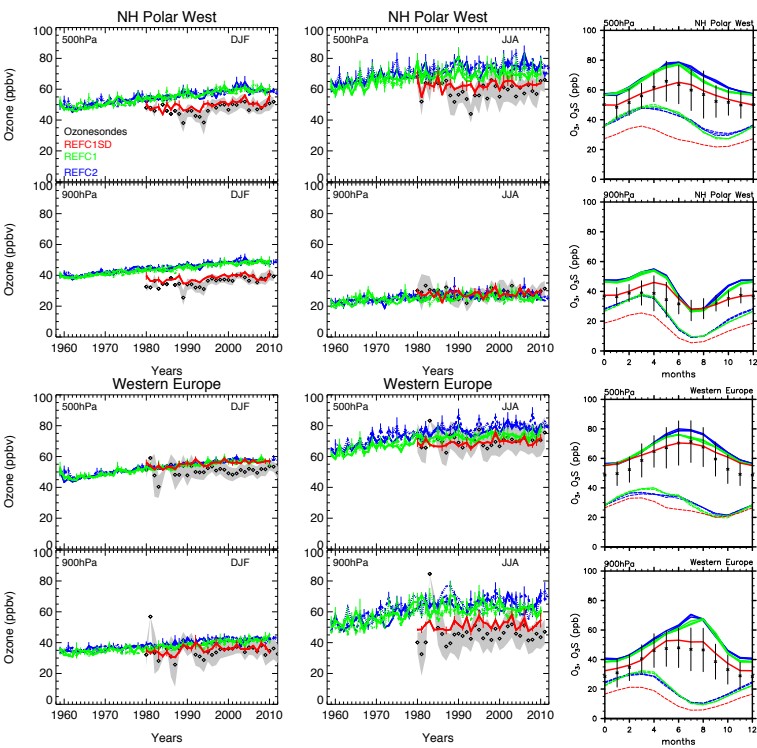

**Figure 3.** Left and middle column: Time evolution of seasonal averaged and regionally aggregated ozone mixing ratios derived from ozone soundings (black diamonds) and model results (colored lines) at two different pressure levels, two different seasons (DJF: left, JJA: right) and regions (NH Polar West, and Western Europe). Grey shading indicates the standard deviation of the observations that include at least 12 observed profiles per year and season. Colored error bars indicate the standard deviation based on monthly-averaged model output. Right column: Regionally aggregated seasonal cycle comparisons of ozone soundings (black lines) and model simulations (colored lines), averaged between 1995 and 2010. Dashed lines indicate mixing ratios of the stratospheric ozone tracer (see text for more details).





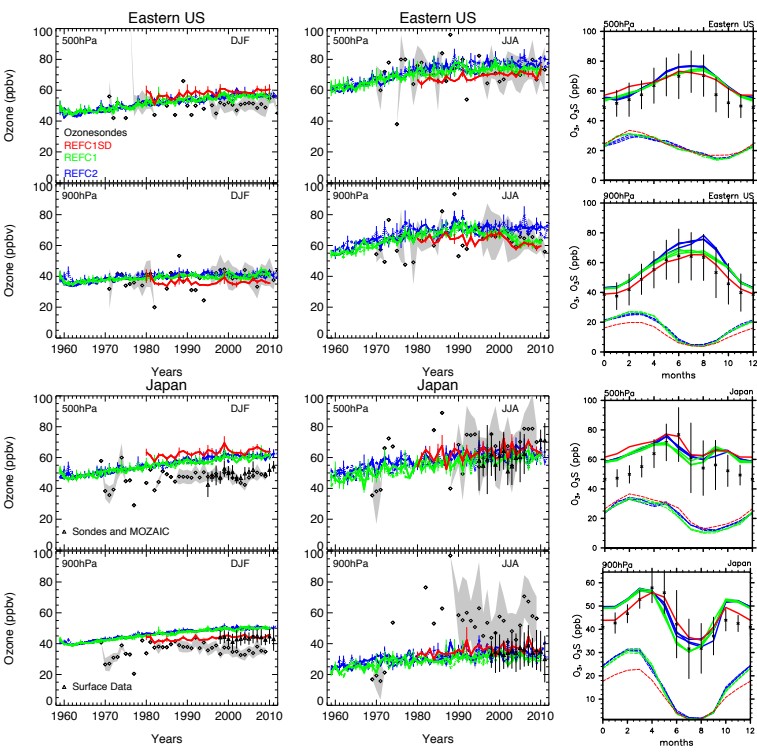

**Figure 4.** As Figure 3, but for Eastern US and Japan instead. For Japan, ozone timeseries compiled by Tanimoto et al. (2015) are added (black triangles) (see text for more details) and used to compare with the seasonal cycle of the model for Japan.





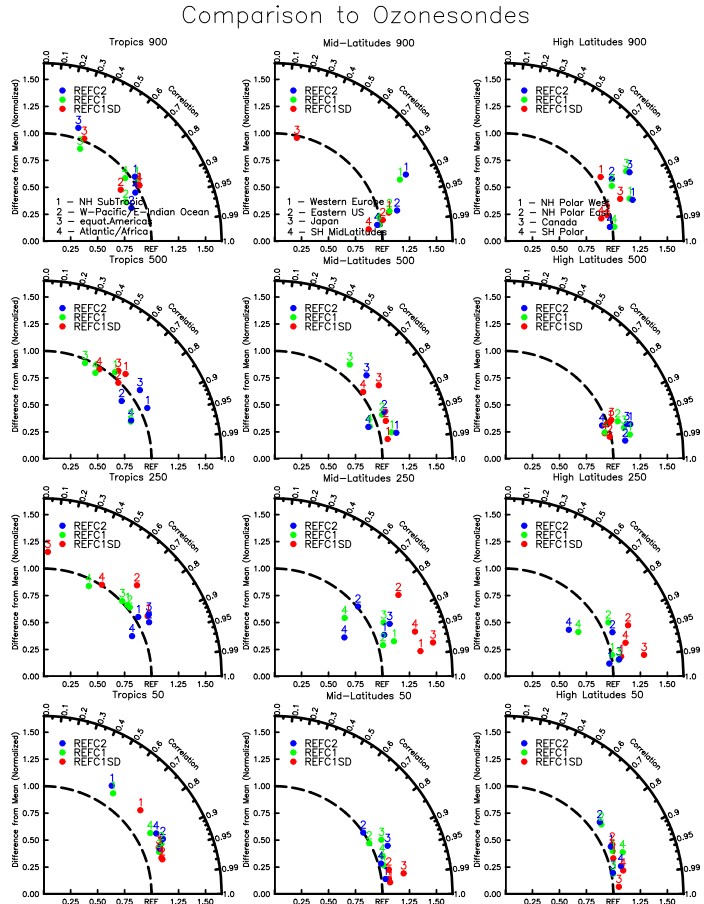

**Figure 5.** Taylor-like diagram comparing the mean and correlation of the seasonal cycle between observations using a present-day ozonesonde climatology between 1995–2011 and model results, interpolated to the same locations as sampled by the observations and for different pressure levels, 900 hPa (top panel), 500 hPa (second panel), 250 hPa (third panel), and 50 hPa (bottom panel). Different numbers correspond to a specific region, as defined in Tilmes et al. (2012). Left panels: 1 – NH-Subtropics; 2 – W-Pacific/East Indian Ocean; 3 – equat. Americas; 4 – Atlantic/Africa. Middle panels: 1 – Western Europe; 2 – Eastern US; 3 – Japan; 4 – SH Mid-Latitudes. Right panels: 1 – NH Polar West; 2 – NH Polar East; 3 – Canada; 4 – SH Polar.




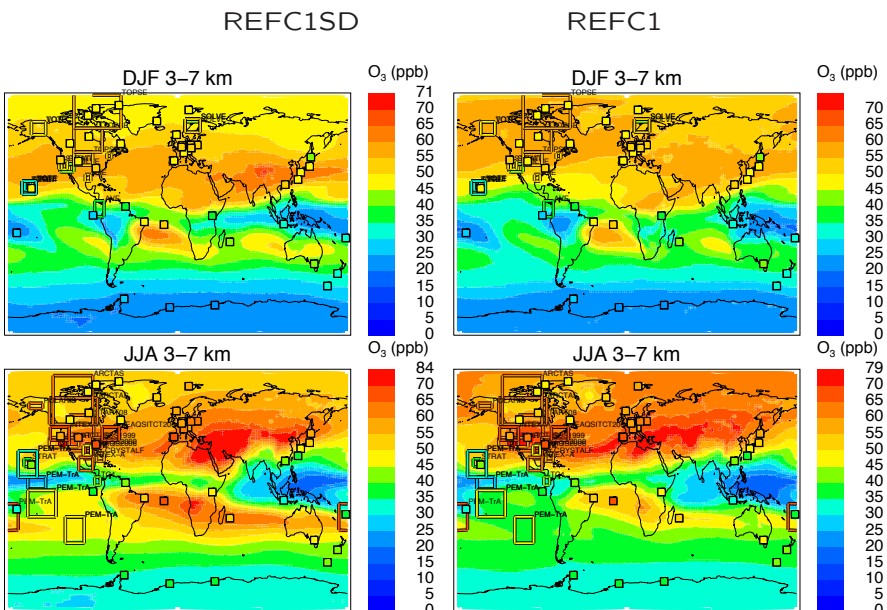

**Figure 6.** Comparison between model results in contours (REFC1SD left and REFC1.1 right) and observations of ozone mixing ratios, averaged over 3-7km for December/January/February (DJF), top, and June/July/August (JJA), bottom. The color of each square represents the value of the observed ozonesonde measurement for the same period and altitude interval, and the color of framed regions corresponds to values derived from aircraft observations averaged over the particular region for each experiment (Tilmes et al., 2015).





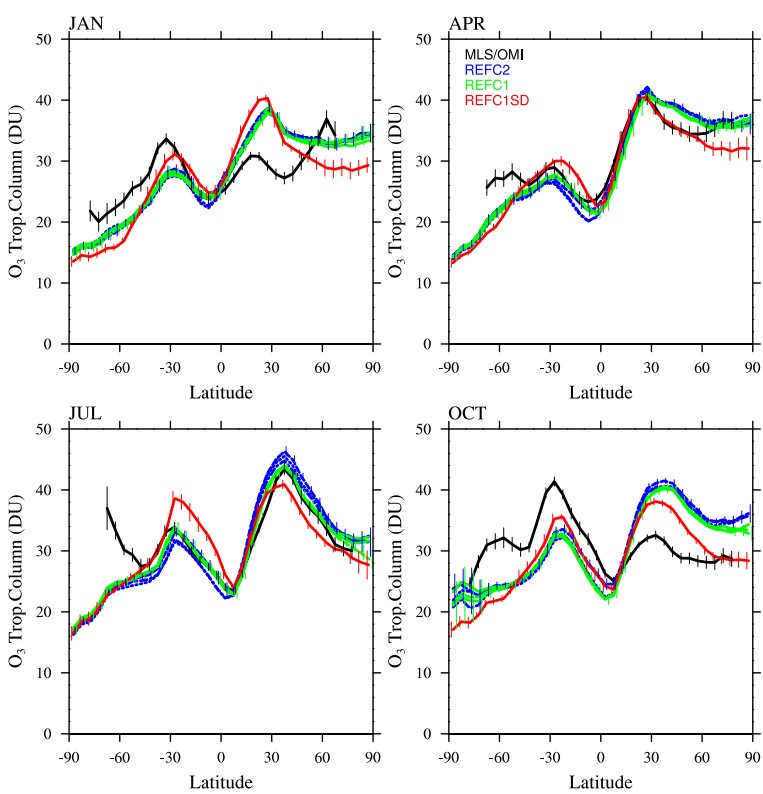

**Figure 7.** Monthly and zonally averaged tropospheric ozone column (in DU) comparison between OMI/MLS observations (black) and different model experiments, see legend, (for ozone < 150 ppb in the model), for four months. Error bars describe the zonally averaged 2 sigma six-year root mean square standard error of the mean at a giving grid point, derived from the $10°$ N to $10°$ S gridded product (Ziemke et al., 2011). Model results are interpolated to the same grid and error bars indicate the standard deviation of the interannual variability per latitude interval.




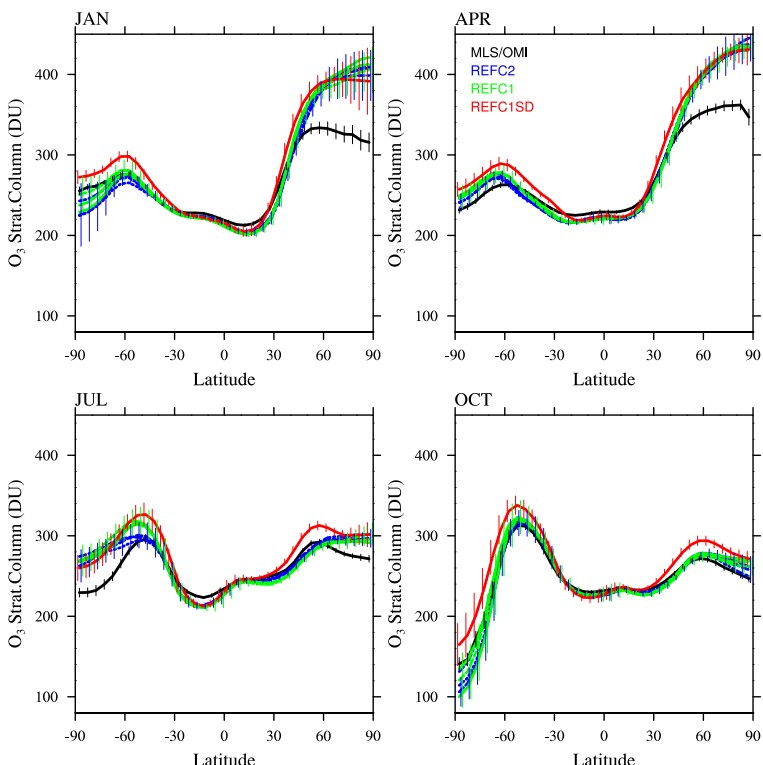

**Figure 8.** As Figure 7, but showing monthly and zonally averaged stratospheric ozone column comparison between OMI/MLS observations (black) and different model experiments, see legend, (for ozone > 150 ppb in the model), for four months.





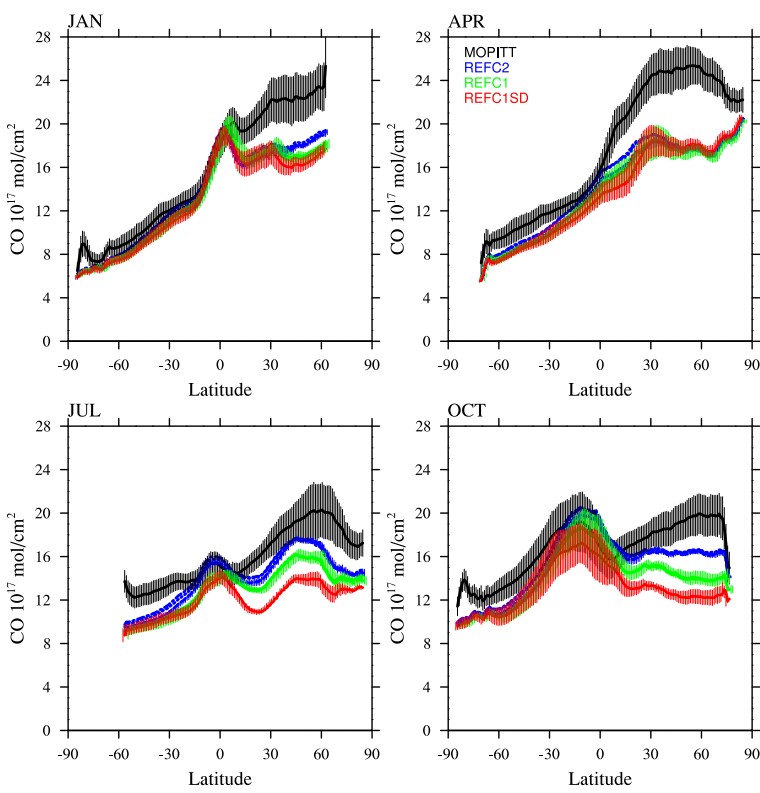

**Figure 9.** Monthly and zonally averaged tropospheric CO column comparison (in molec./cm$^2$) between MOPITT satellite observations (black) and different model experiments, see legend, (for ozone < 150 ppb in the model), for four months. Error bars for observations and model experiments show the standard deviation of the interannual variability per latitude interval.





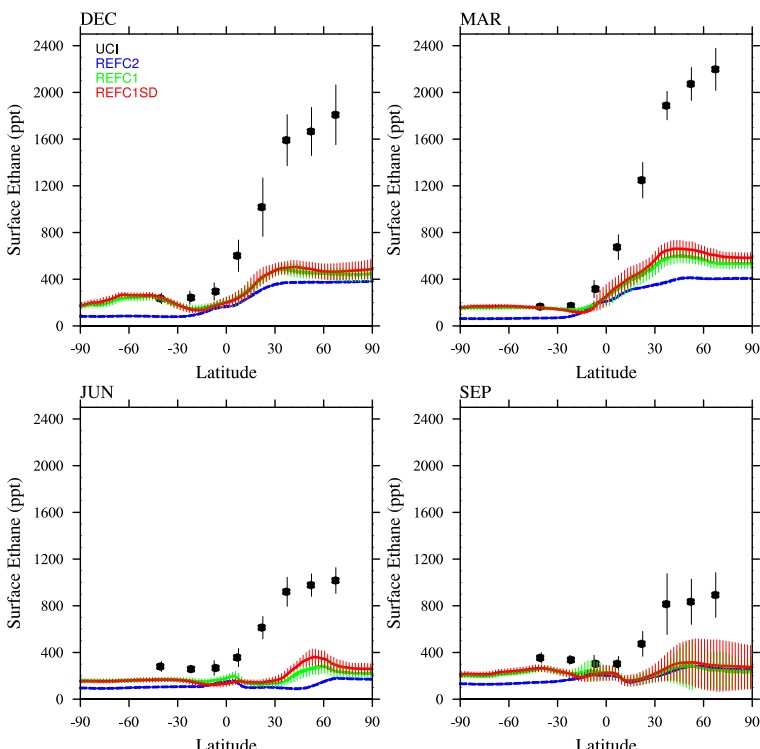

**Figure 10.** Comparison of observed and modeled surface ethane ($C_2H_6$) mixing ratios in each season averaged over 1995-2010 along the length of the Pacific Ocean. Monthly mean CAM4-chem ethane mixing ratios at 190 East are shown for the three model experiments.





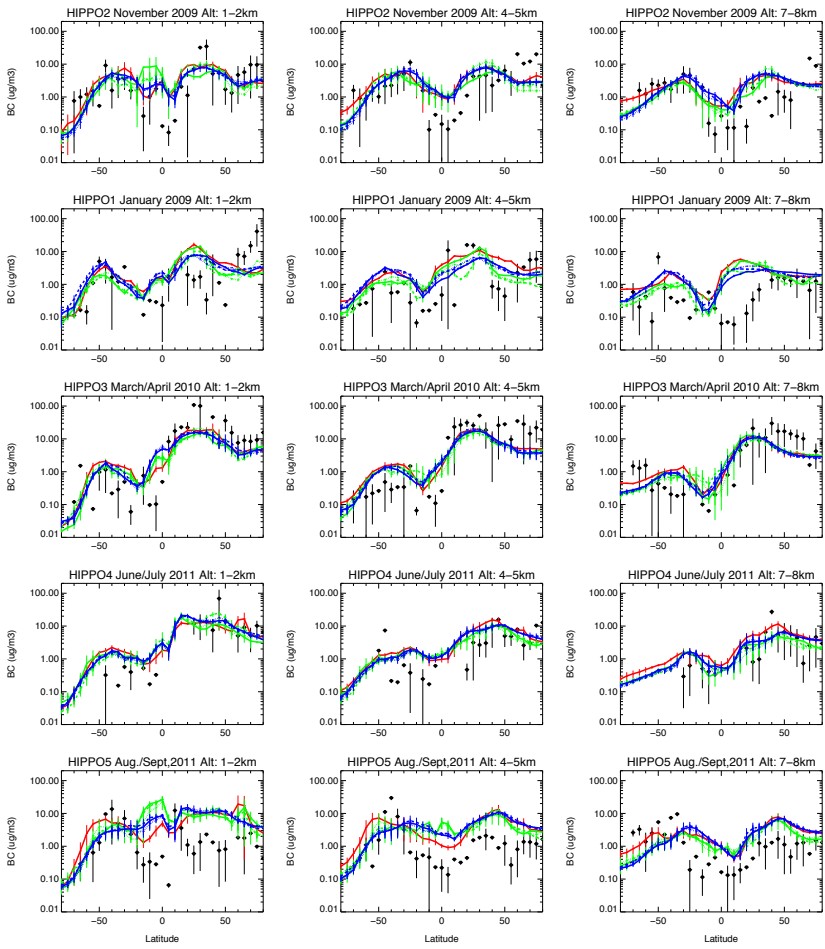

**Figure 11.** Black Carbon comparison between different HIPPO aircraft campaigns taken over the Pacific Ocean (black symbols) and results from the reference simulations REFC1SD (red), REFC1 (green), REFC2 (blue), averaged over different altitude intervals. The sampled aircraft profiles during different HIPPO campaigns were averaged over 5° latitude intervals along the flight path over the Pacific Ocean and compared to model output averaged over the same grid points, as done in Tilmes et al. (2015). The average profiles are averaged over three altitudes regions, 1-2 km, 4-5 km and 7-8 km.





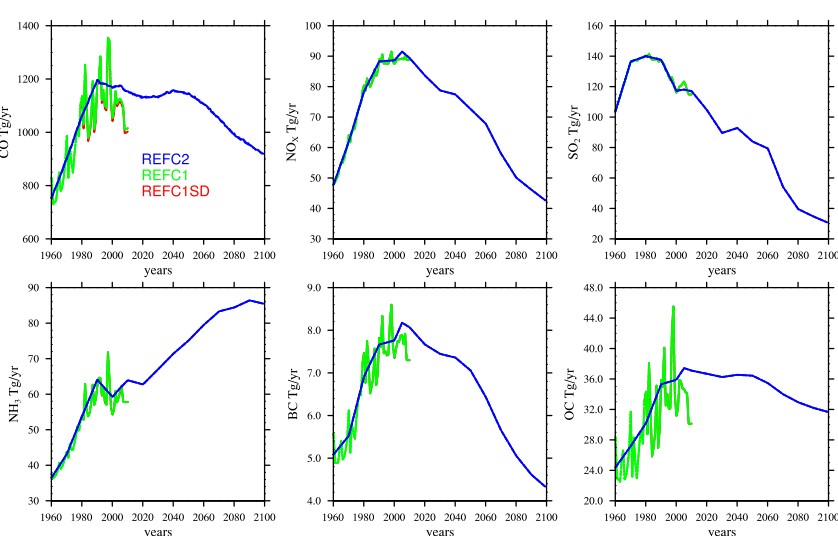

**Figure A1.** Selected surface emissions used for the different reference experiments.





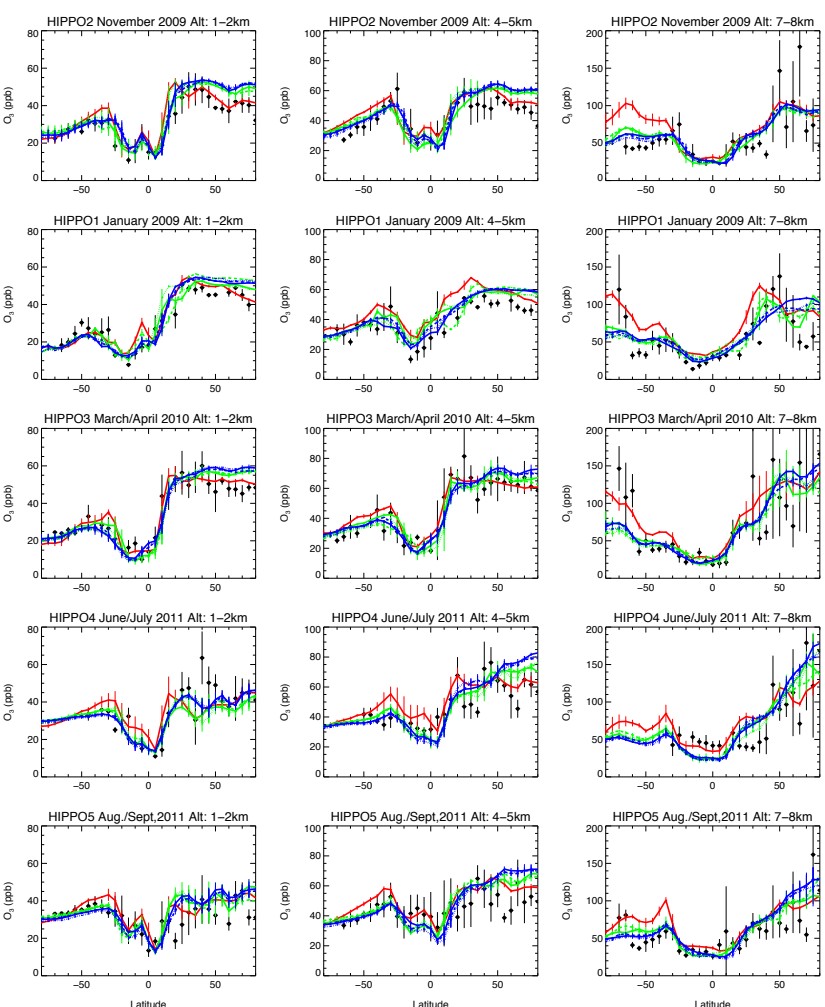

**Figure A2.** O₃ comparison between different HIPPO aircraft campaigns taken over the Pacific Ocean (black symbols) and results from the reference simulations REFC1SD (red), REFC1 (green), REFC2 (blue), averaged over different altitude intervals. The sampled aircraft profiles during different HIPPO campaigns were averaged over 5° latitude intervals along the flight path over the Pacific Ocean and compared to model output averaged over the same grid points, as done in Tilmes et al. (2015). The average profiles are averaged over three altitudes regions, 1-2 km, 4-5 km and 7-8 km.





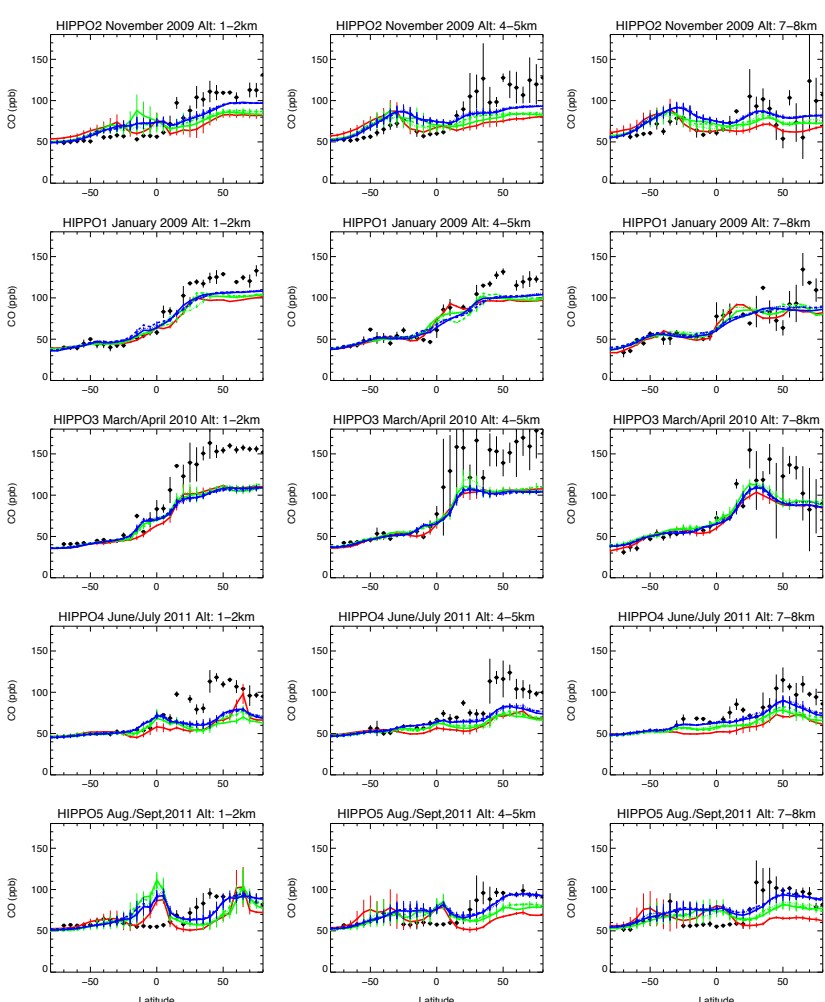

**Figure A3.** As Figure A2, but for Carbon Monoxide.



**Table 1.** Overview of global diagnostics for different experiments, averaged between 1995 and 2010. Lifetimes and burdens are calculated for the troposphere defined for regions where ozone is below 150 ppb.

| CESM1 CAM4chem | REFC1SD | REFC1.1 | REFC1.2 | REFC1.3 | REFC2.1 | REFC2.2 | REFC2.3 |
|---|---|---|---|---|---|---|---|
| Meteorology | MERRA | CAM4 | CAM4 | CAM4 | CAM4 | CAM4 | CAM4 |
| Vert. Res. | 56L | 26L | 26L | 26L | 26L | 26L | 26L |
| TS Global | 288.43 | 288.27 | 288.27 | 288.28 | 288.35 | 288.40 | 288.41 |
| TS Land | 282.37 | 282.10 | 282.12 | 282.17 | 282.23 | 282.20 | 282.23 |
| SWCF | -82.47 | -55.96 | -56.01 | -55.97 | -54.66 | -54.65 | -54.78 |
| $CH_4$ Burden (Tg) | 3991.3 | 4100.5 | 4103.8 | 4099.3 | 4101.4 | 4105.0 | 4103.1 |
| $CH_4$ Lifet. (yr) | 7.6 | 8.0 | 8.1 | 8.1 | 8.2 | 8.2 | 8.2 |
| $CH_3CCl_3$ Lifet. (yr) | 4.5 | 4.8 | 4.8 | 4.8 | 4.9 | 4.9 | 4.9 |
| CO Burden (Tg) | 289.6 | 303.6 | 305.3 | 305.7 | 315.4 | 316.7 | 315.3 |
| CO Emis. (Tg/yr) | 1114.8 | 1119.3 | 1126.5 | 1126.8 | 1170.1 | 1171.1 | 1169.9 |
| CO Dep. (Tg/yr) | 125.8 | 120.7 | 122.0 | 122.1 | 122.7 | 123.0 | 122.9 |
| CO Chem. Loss (Tg/yr) | 2264.1 | 2294.2 | 2295.3 | 2298.0 | 2348.4 | 2353.3 | 2345.5 |
| CO Lifet. (days) | 44.2 | 45.9 | 46.1 | 46.1 | 46.6 | 46.7 | 46.6 |
| $O_3$ Burden (Tg) | 332.5 | 326.9 | 326.5 | 326.4 | 327.8 | 327.2 | 327.0 |
| $O_3$ Dep. (Tg/yr) | 871.7 | 894.4 | 893.9 | 894.2 | 895.0 | 892.8 | 894.7 |
| $O_3$ Chem. Loss (Tg/yr) | 4256.0 | 4268.3 | 4250.6 | 4259.0 | 4287.6 | 4293.5 | 4278.9 |
| $O_3$ Chem. Prod. (Tg/yr) | 4693.8 | 4710.0 | 4706.5 | 4708.3 | 4747.2 | 4756.9 | 4744.1 |
| $O_3$ Net Chem.Change (Tg/yr) | 392.9 | 420.9 | 430.5 | 426.0 | 432.5 | 436.5 | 438.2 |
| $O_3$ STE (Tg/yr) | 478.8 | 473.4 | 463.4 | 468.2 | 462.5 | 456.4 | 456.5 |
| Isop. Emis. (Tg/yr) | 454.2 | 512.6 | 511.8 | 515.0 | 546.6 | 551.6 | 545.6 |
| Monoterp. Emis. (Tg/yr) | 138.9 | 150.0 | 150.0 | 150.3 | 155.4 | 156.4 | 155.0 |
| Methanol Emis. (Tg/yr) | 100.4 | 114.6 | 114.8 | 114.9 | 113.7 | 114.9 | 113.4 |
| Aceton Emis. (Tg/yr) | 41.6 | 44.3 | 44.3 | 44.3 | 47.8 | 48.1 | 47.7 |
| Lightning Prod. (TgN/yr) | 4.5 | 4.8 | 4.7 | 4.8 | 4.7 | 4.7 | 4.7 |
| Total optical depth | 0.107 | 0.119 | 0.119 | 0.119 | 0.118 | 0.118 | 0.118 |
| Dust optical depth | 0.041 | 0.043 | 0.043 | 0.043 | 0.040 | 0.041 | 0.040 |
| POM Burden (TgC) | 0.75 | 0.73 | 0.73 | 0.74 | 0.77 | 0.77 | 0.77 |
| POM Emis. (TgC/yr) | 48.38 | 47.99 | 48.38 | 48.38 | 51.23 | 51.23 | 51.23 |
| POM Lifet. (days) | 7.23 | 7.18 | 7.15 | 7.19 | 7.05 | 7.06 | 7.01 |
| SOA Burden (TgC) | 0.54 | 0.49 | 0.49 | 0.49 | 0.51 | 0.51 | 0.50 |
| SOA Chem. Prod. (TgC/yr) | 32.79 | 34.45 | 34.43 | 34.79 | 35.86 | 36.32 | 35.54 |
| SOA Lifet. (days) | 0.54 | 0.49 | 0.49 | 0.49 | 0.51 | 0.51 | 0.50 |
| BC Burden (TgC) | 0.12 | 0.12 | 0.12 | 0.12 | 0.12 | 0.12 | 0.12 |
| BC Emis. (TgC/yr) | 7.71 | 7.68 | 7.71 | 7.71 | 7.95 | 7.95 | 7.95 |
| BC Lifet. (days) | 7.44 | 7.48 | 7.46 | 7.49 | 5.88 | 5.89 | 5.86 |
| DUST Burden (TgC) | 43.87 | 45.04 | 45.03 | 45.20 | 42.60 | 42.75 | 42.31 |
| SALT Burden (TgC) | 6.02 | 10.88 | 10.88 | 10.87 | 11.14 | 11.10 | 11.11 |
| $SO_4$ Burden (TgS) | 0.45 | 0.49 | 0.49 | 0.49 | 0.51 | 0.51 | 0.51 |
| $SO_4$ Emis. (TgS/yr) | 0.25 | 0.25 | 0.25 | 0.25 | 0.25 | 0.25 | 0.25 |
| $SO_4$ Dry Dep. (TgS/yr) | 5.29 | 5.76 | 5.78 | 5.77 | 5.94 | 6.00 | 5.99 |
| $SO_4$ Wet Dep. (TgS/yr) | -49.93 | -46.36 | -46.28 | -46.30 | -46.36 | -46.42 | -46.49 |
| $SO_4$ Chem. Prod. (TgS/yr) | 10.35 | 10.81 | 10.83 | 10.82 | 10.98 | 11.02 | 11.02 |
| $SO_4$ AQ. Prod. (TgS/yr) | 44.95 | 41.41 | 41.34 | 41.35 | 41.44 | 41.53 | 41.58 |
| $SO_4$ Total Prod. (TgS/yr) | 55.30 | 52.23 | 52.17 | 52.18 | 52.42 | 52.55 | 52.60 |
| $SO_4$ Lifet. (days) | 2.97 | 3.41 | 3.42 | 3.41 | 3.52 | 3.54 | 3.53 |



**Table A1.** Chemical species in CAM4-chem, chemical formula, solver (either explict (E) or semi-implicit (S)), lower boundary conditions (LBC), and wet and dry deposition of species.

| Num. | Species | Formula | Solver | Emis. | LBC | wet dep | dry dep |
|------|---------|---------|--------|-------|-----|---------|---------|
| 1 | ALKO2 | (C5H11O2) | I | | | | |
| 2 | ALKOOH | (C5H12O2) | I | | | X | X |
| 3 | BENO2 | (C6H7O3) | I | | | | |
| 4 | BENOOH | (C6H8O3) | I | | | | |
| 5 | BENZENE | (C6H6) | I | X | | | |
| 6 | BIGALD | (C5H6O2) | I | X | | | |
| 7 | BIGALK | (C5H12) | I | X | | | |
| 8 | BIGENE | (C4H8) | I | X | | | |
| 9 | BR | (Br) | I | | | | |
| 10 | BRCL | (BrCl) | I | | | | |
| 11 | BRO | (BrO) | I | | | | |
| 12 | BRONO2 | (BrONO2) | I | | | X | |
| 13 | BRY | | E | | | | |
| 14 | C10H16 | | I | X | | | |
| 15 | C2H2 | | I | X | | | |
| 16 | C2H4 | | I | X | | | |
| 17 | C2H5O2 | | I | | | | |
| 18 | C2H5OH | | I | X | | X | X |
| 19 | C2H5OOH | | I | | | X | X |
| 20 | C2H6 | | I | X | | | |
| 21 | C3H6 | | I | X | | | |
| 22 | C3H7O2 | | I | | | | |
| 23 | C3H7OOH | | I | | | X | X |
| 24 | C3H8 | | I | X | | | |
| 25 | CCL4 | (CCl4) | E | | X | | |
| 26 | CF2CLBR | (CF2ClBr) | E | | X | | |
| 27 | CF3BR | (CF3Br) | E | | X | | |
| 28 | CFC11 | (CFCl3) | E | | X | | |
| 29 | CFC113 | (CCl2FCClF2) | E | | X | | |
| 30 | CFC114 | (CClF2CClF2) | E | | X | | |
| 31 | CFC115 | (CClF2CF3) | E | | X | | |
| 32 | CFC12 | (CF2Cl2) | E | | X | | |
| 33 | CH2BR2 | (CH2Br2) | E | | X | | |
| 34 | CH2O | | I | X | | X | X |
| 35 | CH3BR | (CH3Br) | E | | X | | |





| Num. | Species | Formula | Solver | Emis. | LBC | wet dep | dry dep |
|---|---|---|---|---|---|---|---|
| 36 | CH3CCL3 | (CH3CCl3) | E | | X | | |
| 37 | CH3CHO | | I | X | | X | X |
| 38 | CH3CL | (CH3Cl) | E | | X | | |
| 39 | CH3CN | | I | X | | X | X |
| 40 | CH3CO3 | | I | | | | |
| 41 | CH3COCH3 | | I | X | | X | X |
| 42 | CH3COCHO | | I | | | X | X |
| 43 | CH3COOH | | I | X | | | X |
| 44 | CH3COOOH | | I | | | X | X |
| 45 | CH3O2 | | I | | | | |
| 46 | CH3OH | | I | X | | X | X |
| 47 | CH3OOH | | I | | | X | X |
| 48 | CH4 | | E | | X | | |
| 49 | CHBR3 | (CHBr3) | E | | X | | |
| 50 | CL | (Cl) | I | | | | |
| 51 | CL2 | (Cl2) | I | | | | |
| 52 | CL2O2 | (Cl2O2) | I | | | | |
| 53 | CLO | (ClO) | I | | | | |
| 54 | CLONO2 | (ClONO2) | I | | | X | |
| 55 | CLY | | E | | | | |
| 56 | CO | | I | X | | | X |
| 57 | CO2 | | E | | X | | |
| 58 | CRESOL | (C7H8O) | I | | | | |
| 59 | DMS | (CH3SCH3) | I | X | | | |
| 60 | ENEO2 | (C4H9O3) | I | | | | |
| 61 | EO | (HOCH2CH2O) | I | | | | |
| 62 | EO2 | (HOCH2CH2O2) | I | | | | |
| 63 | EOOH | (HOCH2CH2OOH) | I | | | X | X |
| 64 | GLYALD | (HOCH2CHO) | I | | | X | X |
| 65 | GLYOXAL | (C2H2O2) | I | | | | |
| 66 | H | | I | | | | |
| 67 | H1202 | (CBr2F2) | E | | X | | |
| 68 | H2 | | I | | X | | |
| 69 | H2402 | (CBrF2CBrF2) | E | | X | | |
| 70 | H2O | | I | | | | |
| 71 | H2O2 | | I | | | X | X |
| 72 | HBR | (HBr) | I | | | X | |
| 73 | HCFC141B | (CH3CCl2F) | E | | X | | |
| 74 | HCFC142B | (CH3CClF2) | E | | X | | |





| Num. | Species | Formula | Solver | Emis. | LBC | wet dep | dry dep |
|------|---------|---------|--------|-------|-----|---------|---------|
| 75 | HCFC22 | (CHF2Cl) | E | | X | | |
| 76 | HCL | (HCl) | I | | | X | |
| 77 | HCN | | I | X | | X | X |
| 78 | HCOOH | | I | X | | X | X |
| 79 | HNO3 | | I | | | X | X |
| 80 | HO2 | | I | | | | |
| 81 | HO2NO2 | | I | | | X | X |
| 82 | HOBR | (HOBr) | I | | | X | |
| 83 | HOCH2OO | | I | | | | |
| 84 | HOCL | (HOCl) | I | | | X | |
| 85 | HYAC | (CH3COCH2OH) | I | | | X | X |
| 86 | HYDRALD | (HOCH2CCH3CHCHO) | I | | | X | X |
| 87 | ISOP | (C5H8) | I | X | | | |
| 88 | ISOPNO3 | (CH2CHCCH3OOCH2ONO2) | I | | | X | |
| 89 | ISOPO2 | (HOCH2COOCH3CHCH2) | I | | | | |
| 90 | ISOPOOH | (HOCH2COOHCH3CHCH2) | I | | | X | X |
| 91 | MACR | (CH2CCH3CHO) | I | | | X | |
| 92 | MACRO2 | (CH3COCHO2CH2OH) | I | | | | |
| 93 | MACROOH | (CH3COCHOOHCH2OH) | I | | | X | X |
| 94 | MCO3 | (CH2CCH3CO3) | I | | | | |
| 95 | MEK | (C4H8O) | I | X | | | |
| 96 | MEKO2 | (C4H7O3) | I | | | | |
| 97 | MEKOOH | (C4H8O3) | I | | | X | X |
| 98 | MPAN | (CH2CCH3CO3NO2) | I | | | | X |
| 99 | MVK | (CH2CHCOCH3) | I | | | X | |
| 100 | N | | I | | | | |
| 101 | N2O | | E | | X | | |
| 102 | N2O5 | | I | | | | |
| 103 | NH3 | | I | X | | X | X |
| 104 | NO | | I | X | | | X |
| 105 | NO2 | | I | | | | X |
| 106 | NO3 | | I | | | | |
| 107 | O | | I | | | | |
| 108 | O1D | (O) | I | | | | |
| 109 | O3 | | I | | | | X |
| 110 | OCLO | (OClO) | I | | | | |
| 111 | OH | | I | | | | |
| 112 | ONIT | (CH3COCH2ONO2) | I | | | X | X |





| Num. | Species | Formula | Solver | Emis. | LBC | wet dep | dry dep |
|---|---|---|---|---|---|---|---|
| 113 | ONITR | (CH2CCH3CHONO2CH2OH) | I | | | X | X |
| 114 | PAN | (CH3CO3NO2) | I | | | | X |
| 115 | PO2 | (C3H6OHO2) | I | | | | |
| 116 | POOH | (C3H6OHOOH) | I | | | X | X |
| 117 | RO2 | (CH3COCH2O2) | I | | | | |
| 118 | ROOH | (CH3COCH2OOH) | I | | | X | X |
| 119 | SF6 | | E | | X | | |
| 120 | SO2 | | I | X | | X | X |
| 121 | SOGB | (C6H7O3) | I | | | X | X |
| 122 | SOGI | (CH3C4H9O4) | I | | | X | X |
| 123 | SOGM | (C10H16O4) | I | | | X | X |
| 124 | SOGT | (C7H9O3) | I | | | X | X |
| 125 | SOGX | (C8H11O3) | I | | | X | X |
| 126 | TERPO2 | (C10H17O3) | I | | | | |
| 127 | TERPOOH | (C10H18O3) | I | | | X | X |
| 128 | TOLO2 | (C7H9O5) | I | | | | |
| 129 | TOLOOH | (C7H10O5) | I | | | X | X |
| 130 | TOLUENE | (C7H8) | I | X | | | |
| 131 | XO2 | (HOCH2COOCH3CHOHCHO) | I | | | | |
| 132 | XOH | (C7H10O6) | I | | | | |
| 133 | XOOH | (HOCH2COOHCH3CHOHCHO) | I | | | X | X |
| 134 | XYLENE | (C8H10) | I | | | | |
| 135 | XYLO2 | (C8H11O3) | I | | | | |
| 136 | XYLOOH | (C8H12O3) | I | | | | |



| Num. | Aerosols | Formula | Solver | Emis. | LBC | wet dep | dry dep |
|---|---|---|---|---|---|---|---|
| 1 | CB1 | (C), hydrophobic BC | I | X | | X | |
| 2 | CB2 | (C) hydrophilic BC | I | X | | X | |
| 3 | NH4 | | I | | | | NH4 |
| 4 | NH4NO3 | | I | | | | X |
| 5 | OC1 | (C), hydrophobic OC | I | X | | | X |
| 6 | OC2 | (C) hydrophilic OC | I | X | | | X |
| 7 | DST01 | (AlSiO5) | I | | | | |
| 8 | DST02 | (AlSiO5) | I | | | | |
| 9 | DST03 | (AlSiO5) | I | | | | |
| 10 | DST04 | (AlSiO5) | I | | | | |
| 11 | SO4 | | I | | | | X |
| 12 | SOAB | (C6H7O3) | I | | | | X |
| 13 | SOAI | (CH3C4H9O4) | I | | | | X |
| 14 | SOAM | (C10H16O4) | I | | | | X |
| 15 | SOAT | (C7H9O3) | I | | | | X |
| 16 | SOAX | (C8H11O3) | I | | | | X |
| 17 | SSLT01 | (NaCl) | I | | | | |
| 18 | SSLT02 | (NaCl) | I | | | | |
| 19 | SSLT03 | (NaCl) | I | | | | |
| 20 | SSLT04 | (NaCl) | I | | | | |

| Num. | Artificial Tracers | Formula | Solver | Emis. | LBC | wet dep | dry dep |
|---|---|---|---|---|---|---|---|
| 1 | $AOA_{NH}$ | (H) | E | | | | |
| 2 | $CO_{25}$ | (CO) | E | X | | | |
| 3 | $CO_{50}$ | (CO) | E | X | | | |
| 4 | E90 | (CO) | E | X | | | |
| 5 | $E90_{NH}$ | (CO) | E | X | | | |
| 6 | $E90_{SH}$ | (CO) | E | X | | | |
| 7 | $NH_5$ | (H) | E | | | | |
| 8 | $NH_{50}$ | (H) | E | | | | |
| 9 | $NH_{50}W$ | (H) | E | | | X | |
| 10 | O3S | (O3) | E | | | | |
| 11 | SF6em | (SF6) | E | X | | | |
| 12 | SO2t | (SO2) | E | | | X | |
| 13 | $ST80_25$ | (H) | E | | | | |





**Table A2.** Chemical reactions in CAM4-chem

| Photolysis |
| --- |
| O2 + hv → 2*O |
| O3 + hv → O1D + O2 |
| O3 + hv → O + O2 |
| N2O + hv → O1D + N2 |
| NO + hv → N + O |
| NO2 + hv → NO + O |
| N2O5 + hv → NO2 + NO3 |
| N2O5 + hv → NO + O + NO3 |
| HNO3 + hv → NO2 + OH |
| NO3 + hv → NO2 + O |
| NO3 + hv → NO + O2 |
| HO2NO2 + hv → OH + NO3 |
| HO2NO2 + hv → NO2 + HO2 |
| CH3OOH + hv → CH2O + H + OH |
| CH2O + hv → CO + 2*H |
| CH2O + hv → CO + H2 |
| H2O + hv → OH + H |
| H2O + hv → H2 + O1D |
| H2O + hv → 2*H + O |
| H2O2 + hv → 2*OH |
| CL2 + hv → 2*CL |
| CLO + hv → CL + O |
| OCLO + hv → O + CLO |
| CL2O2 + hv → 2*CL |
| HOCL + hv → OH + CL |
| HCL + hv → H + CL |
| CLONO2 + hv → CL + NO3 |
| CLONO2 + hv → CLO + NO2 |
| BRCL + hv → BR + CL |
| BRO + hv → BR + O |
| HOBR + hv → BR + OH |
| HBR + hv → BR + H |
| BRONO2 + hv → BR + NO3 |
| BRONO2 + hv → BRO + NO2 |
| CH3CL + hv → CL + CH3O2 |
| CCL4 + hv → 4*CL |
| CH3CCL3 + hv → 3*CL |
| CFC11 + hv → 3*CL |
| CFC12 + hv → 2*CL |
| CFC113 + hv → 3*CL |
| HCFC22 + hv → CL |
| CFC114 + hv → 2*CL |
| CFC115 + hv → CL |




| Photolysis |
| --- |
| HCFC141B + hv → 2*CL |
| HCFC142B + hv → CL |
| CH3BR + hv → BR + CH3O2 |
| CF3BR + hv → BR |
| H1202 + hv → 2*BR |
| H2402 + hv → 2*BR |
| CF2CLBR + hv → BR + CL |
| CHBR3 + hv → 3*BR |
| CH2BR2 + hv → 2*BR |
| CO2 + hv → CO + O |
| CH4 + hv → H + CH3O2 |
| CH4 + hv → 1.44*H2 + 0.18*CH2O + 0.18*O + 0.33*OH + 0.33*H + 0.44*CO2 + 0.38*CO + 0.05*H2O |
| CH3CHO + hv → CH3O2 + CO + HO2 |
| POOH + hv → CH3CHO + CH2O + HO2 + OH |
| CH3COOOH + hv → CH3O2 + OH + CO2 |
| PAN + hv → .6*CH3CO3 + .6*NO2 + .4*CH3O2 + .4*NO3 + .4*CO2 |
| MPAN + hv → MCO3 + NO2 |
| MACR + hv → 1.34*HO2 + .66*MCO3 + 1.34*CH2O + 1.34*CH3CO3 |
| MACR + hv → .66*HO2 + 1.34*CO |
| MVK + hv → .7*C3H6 + .7*CO + .3*CH3O2 + .3*CH3CO3 |
| C2H5OOH + hv → CH3CHO + HO2 + OH |
| EOOH + hv → EO + OH |
| C3H7OOH + hv → 0.82*CH3COCH3 + OH + HO2 |
| ROOH + hv → CH3CO3 + CH2O + OH |
| CH3COCH3 + hv → CH3CO3 + CH3O2 |
| CH3COCHO + hv → CH3CO3 + CO + HO2 |
| XOOH + hv → OH |
| ONITR + hv → HO2 + CO + NO2 + CH2O |
| ISOPOOH + hv → .402*MVK + .288*MACR + .69*CH2O + HO2 |
| HYAC + hv → CH3CO3 + HO2 + CH2O |
| GLYALD + hv → 2*HO2 + CO + CH2O |
| MEK + hv → CH3CO3 + C2H5O2 |
| BIGALD + hv → .45*CO + .13*GLYOXAL + .56*HO2 + .13*CH3CO3 + .18*CH3COCHO |
| GLYOXAL + hv → 2*CO + 2*HO2 |
| ALKOOH + hv → .4*CH3CHO + .1*CH2O + .25*CH3COCH3 + .9*HO2 + .8*MEK + OH |
| MEKOOH + hv → OH + CH3CO3 + CH3CHO |
| TOLOOH + hv → OH + .45*GLYOXAL + .45*CH3COCHO + .9*BIGALD |
| TERPOOH + hv → OH + .1*CH3COCH3 + HO2 + MVK + MACR |
| SF6 + hv → sink |
| SF6em + hv → sink |



| Odd-Oxygen Reactions | Rate |
|---|---|
| O + O2 + M → O3 + M | 6.E-34*(300/T)**2.4 |
| O + O3 → 2*O2 | 8.00E-12*exp( -2060./t) |
| O + O + M → O2 + M | 2.76E-34*exp( 720./t) |

| Odd-Oxygen Reactions (O1D only) | |
|---|---|
| O1D + N2 → O + N2 | 2.15E-11*exp( 110./t) |
| O1D + O2 → O + O2 | 3.30E-11*exp( 55./t) |
| O1D + H2O → 2*OH | 1.63E-10*exp( 60./t) |
| O1D + N2O → 2*NO | 7.25E-11*exp( 20./t) |
| O1D + N2O → N2 + O2 | 4.63E-11*exp( 20./t) |
| O1D + O3 → O2 + O2 | 1.20E-10 |
| O1D + CFC11 → 3*CL | 2.02E-10 |
| O1D + CFC12 → 2*CL | 1.20E-10 |
| O1D + CFC113 → 3*CL | 1.50E-10 |
| O1D + CFC114 → 2*CL | 9.75E-11 |
| O1D + CFC115 → CL | 1.50E-11 |
| O1D + HCFC22 → CL | 7.20E-11 |
| O1D + HCFC141B → 2*CL | 1.79E-10 |
| O1D + HCFC142B → CL | 1.63E-10 |
| O1D + CCL4 → 4*CL | 2.84E-10 |
| O1D + CH3BR → BR | 1.67E-10 |
| O1D + CF2CLBR → CL + BR | 9.60E-11 |
| O1D + CF3BR → BR | 4.10E-11 |
| O1D + H1202 → 2*BR | 1.01E-10 |
| O1D + H2402 → 2*BR | 1.20E-10 |
| O1D + CHBR3 → 3*BR | 4.49E-10 |
| O1D + CH2BR2 → 2*BR | 2.57E-10 |
| O1D + CH4 → CH3O2 + OH | 1.31E-10 |
| O1D + CH4 → CH2O + H + HO2 | 3.50E-11 |
| O1D + CH4 → CH2O + H2 | 9.00E-12 |
| O1D + H2 → H + OH | 1.20E-10 |
| O1D + HCL → CL + OH | 1.50E-10 |
| O1D + HBR → BR + OH | 1.20E-10 |
| O1D + HCN → OH | 7.70E-11*exp( 100./t) |

| Odd Hydrogen Reactions | |
|---|---|
| H + O2 + M → HO2 + M | ko=4.40E-32*(300/t)**1.30 |
| | ki=7.50E-11*(300/t)**-0.20 |
| | f=0.60 |
| H + O3 → OH + O2 | 1.40E-10*exp( -470./t) |
| H + HO2 → 2*OH | 7.20E-11 |
| H + HO2 → H2 + O2 | 6.90E-12 |
| H + HO2 → H2O + O | 1.60E-12 |
| OH + O → H + O2 | 1.80E-11*exp( 180./t) |
| OH + O3 → HO2 + O2 | 1.70E-12*exp( -940./t) |
| OH + HO2 → H2O + O2 | 4.80E-11*exp( 250./t) |
| OH + OH → H2O + O | 1.80E-12 |
| OH + OH + M → H2O2 + M | ko=6.90E-31*(300/t)**1.00 |
| | ki=2.60E-11 |
| | f=0.60 |



| Odd Hydrogen Reactions | |
| --- | --- |
| OH + H2 → H2O + H | 2.80E-12*exp( -1800./t) |
| OH + H2O2 → H2O + HO2 | 1.80E-12 |
| H2 + O → OH + H | 1.60E-11*exp( -4570./t) |
| HO2 + O → OH + O2 | 3.00E-11*exp( 200./t) |
| HO2 + O3 → OH + 2*O2 | 1.00E-14*exp( -490./t) |
| HO2 + HO2 → H2O2 + O2 | 3.0E-13*exp(460/t) |
| | + 2.1E-33 * [M] * exp (920/t)) |
| | * (1 + 1.4E-21 * [H2O] exp (2200/t)) |
| H2O2 + O → OH + HO2 | 1.40E-12*exp( -2000./t) |
| HCN + OH + M → HO2 + M | ko=4.28E-33 |
| | ki=9.30E-15*(300/t)**-4.42 |
| | f=0.80 |
| CH3CN + OH → HO2 | 7.80E-13*exp( -1050./t) |

| Odd Nitrogen Reactions | |
| --- | --- |
| N + O2 → NO + O | 1.50E-11*exp( -3600./t) |
| N + NO → N2 + O | 2.10E-11*exp( 100./t) |
| N + NO2 → N2O + O | 2.90E-12*exp( 220./t) |
| N + NO2 → 2*NO | 1.45E-12*exp( 220./t) |
| N + NO2 → N2 + O2 | 1.45E-12*exp( 220./t) |
| NO + O + M → NO2 + M | ko=9.00E-32*(300/t)**1.50 |
| | ki=3.00E-11 |
| | f=0.60 |
| NO + HO2 → NO2 + OH | 3.30E-12*exp( 270./t) |
| NO + O3 → NO2 + O2 | 3.00E-12*exp( -1500./t) |
| NO2 + O → NO + O2 | 5.10E-12*exp( 210./t) |
| NO2 + O + M → NO3 + M | ko=2.50E-31*(300/t)**1.80 |
| | ki=2.20E-11*(300/t)**0.70 |
| | f=0.60 |
| NO2 + O3 → NO3 + O2 | 1.20E-13*exp( -2450./t) |
| NO2 + NO3 + M → N2O5 + M | ko=2.00E-30*(300/t)**4.40 |
| | ki=1.40E-12*(300/t)**0.70 |
| | f=0.60 |
| N2O5 + M → NO2 + NO3 + M | k(NO2 + NO3 + M) |
| | * 3.704E26 * exp(-11000./t) |
| NO2 + OH + M → HNO3 + M | ko=1.80E-30*(300/t)**3.00 |
| | ki=2.80E-11 |
| | f=0.60 |
| HNO3 + OH → NO3 + H2O | k0 + k3[M]/(1 + k3[M]/k2) |
| | k0 = 2.4E-14*exp(460/t) |
| | k2 = 2.7E-17*exp(2199/t) |
| | k3 = 6.5E-34*exp(1335/t) |
| NO3 + NO → 2*NO2 | 1.50E-11*exp( 170./t) |
| NO3 + O → NO2 + O2 | 1.00E-11 |
| NO3 + OH → HO2 + NO2 | 2.20E-11 |
| NO3 + HO2 → OH + NO2 + O2 | 3.50E-12 |
| NO2 + HO2 + M → HO2NO2 + M | ko=2.00E-31*(300/t)**3.40 |
| | ki=2.90E-12*(300/t)**1.10 |
| | f=0.60 |
| HO2NO2 + OH → H2O + NO2 + O2 | 1.30E-12*exp( 380./t) |
| HO2NO2 + M → HO2 + NO2 + M | k(NO2+HO2+M) |
| | * exp(-10900/t)/2.1E-27 |





| Odd Chlorine Reactions | |
|---|---|
| CL + O3 → CLO + O2 | 2.30E-11*exp( -200./t) |
| CL + H2 → HCL + H | 3.05E-11*exp( -2270./t) |
| CL + H2O2 → HCL + HO2 | 1.10E-11*exp( -980./t) |
| CL + HO2 → HCL + O2 | 1.40E-11*exp( 270./t) |
| CL + HO2 → OH + CLO | 3.60E-11*exp( -375./t) |
| CL + CH2O → HCL + HO2 + CO | 8.10E-11*exp( -30./t) |
| CL + CH4 → CH3O2 + HCL | 7.30E-12*exp( -1280./t) |
| CLO + O → CL + O2 | 2.80E-11*exp( 85./t) |
| CLO + OH → CL + HO2 | 7.40E-12*exp( 270./t) |
| CLO + OH → HCL + O2 | 6.00E-13*exp( 230./t) |
| CLO + HO2 → O2 + HOCL | 2.60E-12*exp( 290./t) |
| CLO + CH3O2 → CL + HO2 + CH2O | 3.30E-12*exp( -115./t) |
| CLO + NO → NO2 + CL | 6.40E-12*exp( 290./t) |
| CLO + NO2 + M → CLONO2 + M | ko=1.80E-31*(300/t)**3.40 |
| | ki=1.50E-11*(300/t)**1.90 |
| | f=0.60 |
| CLO + CLO → 2*CL + O2 | 3.00E-11*exp( -2450./t) |
| CLO + CLO → CL2 + O2 | 1.00E-12*exp( -1590./t) |
| CLO + CLO → CL + OCLO | 3.50E-13*exp( -1370./t) |
| CLO + CLO + M → CL2O2 + M | ko=1.60E-32*(300/t)**4.50 |
| | ki=3.00E-12*(300/t)**2.00 |
| | f=0.60 |
| CL2O2 + M → CLO + CLO + M | k(CLO+CLO+M) / (1.72E-27*exp(8649./t)) |
| HCL + OH → H2O + CL | 1.80E-12*exp( -250./t) |
| HCL + O → CL + OH | 1.00E-11*exp( -3300./t) |
| HOCL + O → CLO + OH | 1.70E-13 |
| HOCL + CL → HCL + CLO | 3.40E-12*exp( -130./t) |
| HOCL + OH → H2O + CLO | 3.00E-12*exp( -500./t) |
| CLONO2 + O → CLO + NO3 | 3.60E-12*exp( -840./t) |
| CLONO2 + OH → HOCL + NO3 | 1.20E-12*exp( -330./t) |
| CLONO2 + CL → CL2 + NO3 | 6.50E-12*exp( 135./t) |

| Odd Bromine Reactions | |
|---|---|
| BR + O3 → BRO + O2 | 1.60E-11*exp( -780./t) |
| BR + HO2 → HBR + O2 | 4.80E-12*exp( -310./t) |
| BR + CH2O → HBR + HO2 + CO | 1.70E-11*exp( -800./t) |
| BRO + O → BR + O2 | 1.90E-11*exp( 230./t) |
| BRO + OH → BR + HO2 | 1.70E-11*exp( 250./t) |
| BRO + HO2 → HOBR + O2 | 4.50E-12*exp( 460./t) |
| BRO + NO → BR + NO2 | 8.80E-12*exp( 260./t) |
| BRO + NO2 + M → BRONO2 + M | ko=5.20E-31*(300/t)**3.20 |
| | ki=6.90E-12*(300/t)**2.90 |
| | f=0.60 |
| BRO + CLO → BR + OCLO | 9.50E-13*exp( 550./t) |
| BRO + CLO → BR + CL + O2 | 2.30E-12*exp( 260./t) |
| BRO + CLO → BRCL + O2 | 4.10E-13*exp( 290./t) |
| BRO + BRO → 2*BR + O2 | 1.50E-12*exp( 230./t) |
| HBR + OH → BR + H2O | 5.50E-12*exp( 200./t) |
| HBR + O → BR + OH | 5.80E-12*exp( -1500./t) |
| HOBR + O → BRO + OH | 1.20E-10*exp( -430./t) |
| BRONO2 + O → BRO + NO3 | 1.90E-11*exp( 215./t) |





| Organic Halogens Reactions with Cl, OH | Rate |
|---|---|
| CH3CL + CL → HO2 + CO + 2*HCL | 2.17E-11*exp( -1130./t) |
| CH3CL + OH → CL + H2O + HO2 | 2.40E-12*exp( -1250./t) |
| CH3CCL3 + OH → H2O + 3*CL | 1.64E-12*exp( -1520./t) |
| HCFC22 + OH → H2O + CL | 1.05E-12*exp( -1600./t) |
| CH3BR + OH → BR + H2O + HO2 | 2.35E-12*exp( -1300./t) |
| CH3BR + CL → HCL + HO2 + BR | 1.40E-11*exp( -1030./t) |
| HCFC141B + OH → 2*CL | 1.25E-12*exp( -1600./t) |
| HCFC142B + OH → CL | 1.30E-12*exp( -1770./t) |
| CH2BR2 + OH → 2*BR + H2O | 2.00E-12*exp( -840./t) |
| CHBR3 + OH → 3*BR | 1.35E-12*exp( -600./t) |
| CH2BR2 + CL → 2*BR + HCL | 6.30E-12*exp( -800./t) |
| CHBR3 + CL → 3*BR + HCL | 4.85E-12*exp( -850./t) |

| C-1 Degradation (Methane, CO, CH2O and derivatives) | |
|---|---|
| CH4 + OH → CH3O2 + H2O | 2.45E-12*exp( -1775./t) |
| CO + OH → CO2 + H | ki = 2.1E09 * (t/300)**6.1 |
| | ko = 1.5E-13 * (t/300)**0.6 |
| | rate=ko/(1+ko/(ki/M)) |
| | *0.6**(1/(1+log10(ko/(ki/M)**2))) |
| CO + OH + M → CO2 + HO2 + M | ko=5.90E-33*(300/t)**1.40 |
| | ki=1.10E-12*(300/t)**-1.30 |
| | f=0.60 |
| CH2O + NO3 → CO + HO2 + HNO3 | 6.00E-13*exp( -2058./t) |
| CH2O + OH → CO + H2O + H | 5.50E-12*exp( 125./t) |
| CH2O + O → HO2 + OH + CO | 3.40E-11*exp( -1600./t) |
| CH2O + HO2 → HOCH2OO | 9.70E-15*exp( 625./t) |
| CH3O2 + NO → CH2O + NO2 + HO2 | 2.80E-12*exp( 300./t) |
| CH3O2 + HO2 → CH3OOH + O2 | 4.10E-13*exp( 750./t) |
| CH3O2 + CH3O2 → 2*CH2O + 2*HO2 | 5.00E-13*exp( -424./t) |
| CH3O2 + CH3O2 → CH2O + CH3OH | 1.90E-14*exp( 706./t) |
| CH3OH + OH → HO2 + CH2O | 2.90E-12*exp( -345./t) |
| CH3OOH + OH → .7*CH3O2 + .3*OH + .3*CH2O + H2O | 3.80E-12*exp( 200./t) |
| HCOOH + OH → HO2 + CO2 + H2O | 4.50E-13 |
| HOCH2OO → CH2O + HO2 | 2.40E+12*exp( -7000./t) |
| HOCH2OO + NO → HCOOH + NO2 + HO2 | 2.60E-12*exp( 265./t) |
| HOCH2OO + HO2 → HCOOH | 7.50E-13*exp( 700./t) |

| C-2 Degradation | |
|---|---|
| C2H2 + CL + M → CL + M | ko=5.20E-30*(300/t)**2.40 |
| | ki=2.20E-10*(300/t)**0.70 |
| | f=0.60 |
| C2H4 + CL + M → CL + M | ko=1.60E-29*(300/t)**3.30 |
| | ki=3.10E-10*(300/t) |
| | f=0.60 |
| C2H6 + CL → HCL + C2H5O2 | 7.20E-11*exp( -70./t) |
| C2H2 + OH + M → .65*GLYOXAL + .65*OH + .35*HCOOH + .35*HO2 + .35*CO + M | ko=5.50E-30 |
| | ki=8.30E-13*(300/t)**-2.00 |
| | f=0.60 |
| C2H6 + OH → C2H5O2 + H2O | 7.66E-12*exp( -1020./t) |
| C2H4 + OH + M → EO2 + M | ko=8.60E-29*(300/t)**3.10 |
| | ki=9.00E-12*(300/t)**0.85 |
| | f=0.48 |



| C-2 Degradation | |
| --- | --- |
| EO2 + NO → 0.5*CH2O + 0.25*HO2 + 0.75*EO + NO2 | 4.20E-12*exp( 180./t) |
| EO2 + HO2 → EOOH | 7.50E-13*exp( 700./t) |
| EO + O2 → GLYALD + HO2 | 1.00E-14 |
| EO → 2*CH2O + HO2 | 1.60E+11*exp( -4150./t) |
| C2H4 + O3 → CH2O + .12*HO2 + .5*CO + .12*OH + .5*HCOOH | 1.20E-14*exp( -2630./t) |
| CH3COOH + OH → CH3O2 + CO2 + H2O | 7.00E-13 |
| C2H5O2 + NO → CH3CHO + HO2 + NO2 | 2.60E-12*exp( 365./t) |
| C2H5O2 + HO2 → C2H5OOH + O2 | 7.50E-13*exp( 700./t) |
| C2H5O2 + CH3O2 → .7*CH2O + .8*CH3CHO + HO2 + .3*CH3OH + .2*C2H5OH | 2.00E-13 |
| C2H5O2 + C2H5O2 → 1.6*CH3CHO + 1.2*HO2 + .4*C2H5OH | 6.80E-14 |
| C2H5OOH + OH → .5*C2H5O2 + .5*CH3CHO + .5*OH | 3.80E-12*exp( 200./t) |
| CH3CHO + OH → CH3CO3 + H2O | 4.63E-12*exp( 350./t) |
| CH3CHO + NO3 → CH3CO3 + HNO3 | 1.40E-12*exp( -1900./t) |
| CH3CO3 + NO → CH3O2 + CO2 + NO2 | 8.10E-12*exp( 270./t) |
| CH3CO3 + NO2 + M → PAN + M | ko=9.70E-29*(300/t)**5.60 ki=9.30E-12*(300/t)**1.50 f=0.60 |
| CH3CO3 + HO2 → .75*CH3COOOH + .25*CH3COOH + .25*O3 | 4.30E-13*exp( 1040./t) |
| CH3CO3 + CH3O2 → .9*CH3O2 + CH2O + .9*HO2 + .9*CO2 + .1*CH3COOH | 2.00E-12*exp( 500./t) |
| CH3CO3 + CH3CO3 → 2*CH3O2 + 2*CO2 | 2.50E-12*exp( 500./t) |
| CH3COOOH + OH → .5*CH3CO3 + .5*CH2O + .5*CO2 + H2O | 1.00E-12 |
| GLYALD + OH → HO2 + .2*GLYOXAL + .8*CH2O + .8*CO2 | 1.00E-11 |
| GLYOXAL + OH → HO2 + CO + CO2 | 1.15E-11 |
| C2H5OH + OH → HO2 + CH3CHO | 6.90E-12*exp( -230./t) |
| PAN + M → CH3CO3 + NO2 + M | k(CH3CO3+NO2+M) *1.111E28 * exp(-14000/t) |
| PAN + OH → CH2O + NO3 | 4.00E-14 |

| C-3 Degradation | Rate |
| --- | --- |
| C3H6 + OH + M → PO2 + M | ko=8.00E-27*(300/t)**3.50 ki=3.00E-11 f=0.50 |
| C3H6 + O3 → .54*CH2O + .19*HO2 + .33*OH + .08*CH4 + .56*CO + .5*CH3CHO + .31*CH3O2 + .25*CH3COOH | 6.50E-15*exp( -1900./t) |
| C3H6 + NO3 → ONIT | 4.60E-13*exp( -1156./t) |
| C3H7O2 + NO → .82*CH3COCH3 + NO2 + HO2 + .27*CH3CHO | 4.20E-12*exp( 180./t) |
| C3H7O2 + HO2 → C3H7OOH + O2 | 7.50E-13*exp( 700./t) |
| C3H7O2 + CH3O2 → CH2O + HO2 + .82*CH3COCH3 | 3.75E-13*exp( -40./t) |
| C3H7OOH + OH → H2O + C3H7O2 | 3.80E-12*exp( 200./t) |
| C3H8 + OH → C3H7O2 + H2O | 8.70E-12*exp( -615./t) |
| PO2 + NO → CH3CHO + CH2O + HO2 + NO2 | 4.20E-12*exp( 180./t) |
| PO2 + HO2 → POOH + O2 | 7.50E-13*exp( 700./t) |
| POOH + OH → .5*PO2 + .5*OH + .5*HYAC + H2O | 3.80E-12*exp( 200./t) |
| CH3COCH3 + OH → RO2 + H2O | 3.82E-11*exp(-2000/t) + .33E-13 |
| RO2 + NO → CH3CO3 + CH2O + NO2 | 2.90E-12*exp( 300./t) |
| RO2 + HO2 → ROOH + O2 | 8.60E-13*exp( 700./t) |
| RO2 + CH3O2 → .3*CH3CO3 + .8*CH2O + .3*HO2 + .2*HYAC + .5*CH3COCHO + .5*CH3OH | 7.10E-13*exp( 500./t) |





| C-3 Degradation | Rate |
|---|---|
| ROOH + OH → RO2 + H2O | 3.80E-12*exp( 200./t) |
| HYAC + OH → CH3COCHO + HO2 | 3.00E-12 |
| CH3COCHO + OH → CH3CO3 + CO + H2O | 8.40E-13*exp( 830./t) |
| CH3COCHO + NO3 → HNO3 + CO + CH3CO3 | 1.40E-12*exp( -1860./t) |
| ONIT + OH → NO2 + CH3COCHO | 6.80E-13 |

| C-4 Degradation | |
|---|---|
| BIGENE + OH → ENEO2 | 5.40E-11 |
| ENEO2 + NO → CH3CHO + .5*CH2O + .5*CH3COCH3 + HO2 + NO2 | 4.20E-12*exp( 180./t) |
| MVK + OH → MACRO2 | 4.13E-12*exp( 452./t) |
| MVK + O3 → .8*CH2O + .95*CH3COCHO + .08*OH + .2*O3 + .06*HO2 + .05*CO + .04*CH3CHO | 7.52E-16*exp( -1521./t) |
| MEK + OH → MEKO2 | 2.30E-12*exp( -170./t) |
| MEKO2 + NO → CH3CO3 + CH3CHO + NO2 | 4.20E-12*exp( 180./t) |
| MEKO2 + HO2 → MEKOOH | 7.50E-13*exp( 700./t) |
| MEKOOH + OH → MEKO2 | 3.80E-12*exp( 200./t) |
| MACR + OH → .5*MACRO2 + .5*H2O + .5*MCO3 | 1.86E-11*exp( 175./t) |
| MACR + O3 → .8*CH3COCHO + .275*HO2 + .2*CO + .2*O3 + .7*CH2O + .215*OH | 4.40E-15*exp( -2500./t) |
| MACRO2 + NO → NO2 + .47*HO2 + .25*CH2O + .53*GLYALD + .25*CH3COCHO + .53*CH3CO3 + .22*HYAC + .22*CO | 2.70E-12*exp( 360./t) |
| MACRO2 + NO → 0.8*ONITR | 1.30E-13*exp( 360./t) |
| MACRO2 + NO3 → NO2 + .47*HO2 + .25*CH2O + .25*CH3COCHO + .22*CO + .53*GLYALD + .22*HYAC + .53*CH3CO3 | 2.40E-12 |
| MACRO2 + HO2 → MACROOH | 8.00E-13*exp( 700./t) |
| MACRO2 + CH3O2 → .73*HO2 + .88*CH2O + .11*CO + .24*CH3COCHO + .26*GLYALD + .26*CH3CO3 + .25*CH3OH + .23*HYAC | 5.00E-13*exp( 400./t) |
| MACRO2 + CH3CO3 → .25*CH3COCHO + CH3O2 + .22*CO + .47*HO2 + .53*GLYALD + .22*HYAC + .25*CH2O + .53*CH3CO3 | 1.40E-11 |
| MACROOH + OH → .5*MCO3 + .2*MACRO2 + .1*OH + .2*HO2 | 2.30E-11*exp( 200./t) |
| MCO3 + NO → NO2 + CH2O + CH3CO3 | 5.30E-12*exp( 360./t) |
| MCO3 + NO3 → NO2 + CH2O + CH3CO3 | 5.00E-12 |
| MCO3 + HO2 → .25*O3 + .25*CH3COOH + .75*CH3COOOH + .75*O2 | 4.30E-13*exp( 1040./t) |
| MCO3 + CH3O2 → 2*CH2O + HO2 + CO2 + CH3CO3 | 2.00E-12*exp( 500./t) |
| MCO3 + CH3CO3 → 2*CO2 + CH3O2 + CH2O + CH3CO3 | 4.60E-12*exp( 530./t) |
| MCO3 + MCO3 → 2*CO2 + 2*CH2O + 2*CH3CO3 | 2.30E-12*exp( 530./t) |
| MCO3 + NO2 + M → MPAN + M | 1.1E-11*300./t/[M] |
| MPAN + M → MCO3 + NO2 + M | k(MCO3 + NO2 + M) * 1.111E28 * exp(-14000/t) |
| MPAN + OH + M → .5*HYAC + .5*NO3 + .5*CH2O + .5*HO2 + 0.5*CO2 + M | ko=8.00E-27*(300/t)**3.50 ki=3.00E-11 f=0.50 |



| C-5 Degradation | |
| --- | --- |
| ISOP + OH → ISOPO2 | 2.54E-11*exp( 410./t) |
| ISOP + O3 → .4*MACR + .2*MVK + .07*C3H6 + .27*OH + .06*HO2 + .6*CH2O + .3*CO + .1*O3 + .2*MCO3 + .2*CH3COOH | 1.05E-14*exp( -2000./t) |
| ISOP + NO3 → ISOPNO3 | 3.03E-12*exp( -446./t) |
| ISOPO2 + NO → .08*ONITR + .92*NO2 + .23*MACR + .32*MVK + .33*HYDRALD + .02*GLYOXAL + .02*GLYALD + .02*CH3COCHO + .02*HYAC + .55*CH2O + .92*HO2 | 4.40E-12*exp( 180./t) |
| ISOPO2 + NO3 → HO2 + NO2 + .6*CH2O + .25*MACR + .35*MVK + .4*HYDRALD | 2.40E-12 |
| ISOPO2 + HO2 → ISOPOOH | 8.00E-13*exp( 700./t) |
| ISOPOOH + OH → .8*XO2 + .2*ISOPO2 | 1.52E-11*exp( 200./t) |
| ISOPO2 + CH3O2 → .25*CH3OH + HO2 + 1.2*CH2O + .19*MACR + .26*MVK + .3*HYDRALD | 5.00E-13*exp( 400./t) |
| ISOPO2 + CH3CO3 → CH3O2 + HO2 + .6*CH2O + .25*MACR + .35*MVK + .4*HYDRALD | 1.40E-11 |
| ISOPNO3 + NO → 1.206*NO2 + .794*HO2 + .072*CH2O + .167*MACR + .039*MVK + .794*ONITR | 2.70E-12*exp( 360./t) |
| ISOPNO3 + NO3 → 1.206*NO2 + .072*CH2O + .167*MACR + .039*MVK + .794*ONITR + .794*HO2 | 2.40E-12 |
| ISOPNO3 + HO2 → .206*NO2 + .206*CH2O + .206*OH + .167*MACR + .039*MVK + .794*ONITR | 8.00E-13*exp( 700./t) |
| BIGALK + OH → ALKO2 | 3.50E-12 |
| ONITR + OH → HYDRALD + .4*NO2 + HO2 | 4.50E-11 |
| ONITR + NO3 → HO2 + NO2 + HYDRALD | 1.40E-12*exp( -1860./t) |
| HYDRALD + OH → XO2 | 1.86E-11*exp( 175./t) |
| ALKO2 + NO → .4*CH3CHO + .1*CH2O + .25*CH3COCH3 + .9*HO2 + .8*MEK + .9*NO2 + .1*ONIT | 4.20E-12*exp( 180./t) |
| ALKO2 + HO2 → ALKOOH | 7.50E-13*exp( 700./t) |
| ALKOOH + OH → ALKO2 | 3.80E-12*exp( 200./t) |
| XO2 + NO → NO2 + HO2 + .25*CO + .25*CH2O + .25*GLYOXAL + .25*CH3COCHO + .25*HYAC + .25*GLYALD | 2.70E-12*exp( 360./t) |
| XO2 + NO3 → NO2 + HO2 + 0.5*CO + .25*HYAC + 0.25*GLYOXAL + .25*CH3COCHO + .25*GLYALD | 2.40E-12 |
| XO2 + HO2 → XOOH | 8.00E-13*exp( 700./t) |
| XO2 + CH3O2 → .3*CH3OH + .8*HO2 + .8*CH2O + .2*CO + .1*GLYOXAL + .1*CH3COCHO + .1*HYAC + .1*GLYALD | 5.00E-13*exp( 400./t) |
| XO2 + CH3CO3 → .25*CO + .25*CH2O + .25*GLYOXAL + CH3O2 + HO2 + .25*CH3COCHO + .25*HYAC + .25*GLYALD + CO2 | 1.30E-12*exp( 640./t) |
| XOOH + OH → H2O + XO2 | 1.90E-12*exp( 190./t) |
| XOOH + OH → H2O + OH | T**2 * 7.69E-17 * exp(253./t) |




| C-7 Degradation | Rate |
|---|---|
| TOLUENE + OH → .25*CRESOL + .25*HO2 + .7*TOLO2 | 1.70E-12*exp( 352./t) |
| TOLO2 + NO → .45*GLYOXAL + .45*CH3COCHO + .9*BIGALD + .9*NO2 + .9*HO2 | 4.20E-12*exp( 180./t) |
| TOLO2 + HO2 → TOLOOH | 7.50E-13*exp( 700./t) |
| TOLOOH + OH → TOLO2 | 3.80E-12*exp( 200./t) |
| CRESOL + OH → XOH | 3.00E-12 |
| XOH + NO2 → .7*NO2 + .7*BIGALD + .7*HO2 | 1.00E-11 |
| BENZENE + OH → BENO2 | 2.30E-12*exp( -193./t) |
| BENO2 + HO2 → BENOOH | 1.40E-12*exp( 700./t) |
| BENO2 + NO → 0.9*GLYOXAL + 0.9*BIGALD + 0.9*NO2 + 0.9*HO2 | 2.60E-12*exp( 350./t) |
| XYLENE + OH → XYLO2 | 2.30E-11 |
| XYLO2 + HO2 → XYLOOH | 1.40E-12*exp( 700./t) |
| XYLO2 + NO → 0.62*BIGALD + 0.34*GLYOXAL + 0.54*CH3COCHO + 0.9*NO2 + 0.9*HO2 | 2.60E-12*exp( 350./t) |

| C-10 Degradation | |
|---|---|
| C10H16 + OH → TERPO2 | 1.20E-11*exp( 444./t) |
| C10H16 + O3 → .7*OH + MVK + MACR + HO2 | 1.00E-15*exp( -732./t) |
| C10H16 + NO3 → TERPO2 + NO2 | 1.20E-12*exp( 490./t) |
| TERPO2 + NO → .1*CH3COCH3 + HO2 + MVK + MACR + NO2 | 4.20E-12*exp( 180./t) |
| TERPO2 + HO2 → TERPOOH | 7.50E-13*exp( 700./t) |
| TERPOOH + OH → TERPO2 | 3.80E-12*exp( 200./t) |

| Tropospheric Heterogeneous Reactions | |
|---|---|
| N2O5 → 2*HNO3 | |
| NO3 → HNO3 | |
| NO2 → 0.5*OH + 0.5*NO + 0.5*HNO3 | |
| CB1 → CB2 | 7.10E-06 |
| SO2 + OH → SO4 | |
| DMS + OH → SO2 | 9.60E-12*exp( -234./t) |
| DMS + OH → .5*SO2 + .5*HO2 | |
| DMS + NO3 → SO2 + HNO3 | 1.90E-13*exp( 520./t) |
| NH3 + OH → H2O | 1.70E-12*exp( -710./t) |
| OC1 → OC2 | 7.10E-06 |
| HO2 → 0.5*H2O2 | |

| Stratospheric removal rates for BAM aerosols | |
|---|---|
| CB1 → (No products) | 6.34E-08 |
| CB2 → (No products) | 6.34E-08 |
| OC1 → (No products) | 6.34E-08 |
| OC2 → (No products) | 6.34E-08 |
| SO4 → (No products) | 6.34E-08 |





| Stratospheric removal rates for BAM aerosols | |
|---|---|
| SOAM → (No products) | 6.34E-08 |
| SOAI → (No products) | 6.34E-08 |
| SOAB → (No products) | 6.34E-08 |
| SOAT → (No products) | 6.34E-08 |
| SOAX → (No products) | 6.34E-08 |
| NH4 → (No products) | 6.34E-08 |
| NH4NO3 → (No products) | 6.34E-08 |
| SSLT01 → (No products) | 6.34E-08 |
| SSLT02 → (No products) | 6.34E-08 |
| SSLT03 → (No products) | 6.34E-08 |
| SSLT04 → (No products) | 6.34E-08 |
| DST01 → (No products) | 6.34E-08 |
| DST02 → (No products) | 6.34E-08 |
| DST03 → (No products) | 6.34E-08 |
| DST04 → (No products) | 6.34E-08 |
| SO2t → (No products) | 6.34E-08 |
| **Sulfate aerosol reactions** | |
| N2O5 → 2*HNO3 | f (sulfuric acid wt%) |
| CLONO2 → HOCL + HNO3 | f (T,P,HCl,H2O,r) |
| BRONO2 → HOBR + HNO3 | f (T,P,H2O,r) |
| CLONO2 + HCL → CL2 + HNO3 | f (T,P,HCl,H2O,r) |
| HOCL + HCL → CL2 + H2O | f (T,P,HCl,HOCl,H2O,r) |
| HOBR + HCL → BRCL + H2O | f (T,P,HCl,HOBr,H2O,r) |
| **Nitric acid Di-hydrate reactions** | |
| N2O5 → 2*HNO3 | $\gamma = 0.0004$ |
| CLONO2 → HOCL + HNO3 | $\gamma = 0.004$ |
| CLONO2 + HCL → CL2 + HNO3 | $\gamma = 0.2$ |
| HOCL + HCL → CL2 + H2O | $\gamma = 0.1$ |
| BRONO2 → HOBR + HNO3 | $\gamma = 0.3$ |
| **Ice aerosol reactions** | |
| N2O5 → 2*HNO3 | $\gamma = 0.02$ |
| CLONO2 → HOCL + HNO3 | $\gamma = 0.3$ |
| BRONO2 → HOBR + HNO3 | $\gamma = 0.3$ |
| CLONO2 + HCL → CL2 + HNO3 | $\gamma = 0.3$ |
| HOCL + HCL → CL2 + H2O | $\gamma = 0.2$ |
| HOBR + HCL → BRCL + H2O | $\gamma = 0.3$ |
| **Synthetic tracer reactions** | |
| $NH_5$ → (No products) | 2.31E-06 |
| $NH_{50}$ → (No products) | 2.31E-07 |
| $NH_{50W}$ → (No products) | 2.31E-07 |
| $ST80_{25}$ → (No products) | 4.63E-07 |
| $CO_{25}$ → (No products) | 4.63E-07 |
| $CO_{50}$ → (No products) | 2.31E-07 |
| E90 → (No products) | 1.29E-07 |
| $E90_{NH}$ → (No products) | 1.29E-07 |
| $E90_{SH}$ → (No products) | 1.29E-07 |



**Table A3.** Tropospheric ozone production and loss rates calculated for explict reaction rates, O3-Prod and O3-Loss are the sum of the specific reaction rates.

| Production / Loss (Tg/yr) | REFC1SD | REFC1 | REFC2 |
|---|---|---|---|
| O3-Prod | 4701.1 | 4716.5 | 4758.0 |
| NO-HO2 | 3032.2 | 3017.3 | 3051.7 |
| CH3O2-NO | 1102.1 | 1078.6 | 1072.2 |
| PO2-NO | 19.8 | 20.9 | 21.1 |
| CH3CO3-NO | 159.6 | 168.8 | 172.3 |
| C2H5O2-NO | 8.2 | 8.1 | 7.5 |
| 0.92*ISOPO2-NO | 113.0 | 131.8 | 136.1 |
| MACRO2-NOa | 60.9 | 68.3 | 69.9 |
| MCO3-NO | 25.6 | 28.9 | 29.8 |
| C3H7O2-NO | n.a. | n.a. | n.a. |
| RO2-NO | 10.6 | 11.2 | 11.6 |
| XO2-NO | 53.6 | 62.4 | 64.1 |
| 0.9*TOLO2-NO | 2.7 | 2.8 | 3.8 |
| TERPO2-NO | 15.2 | 16.7 | 16.8 |
| 0.9*ALKO2-NO | 21.6 | 21.3 | 21.7 |
| ENEO2-NO | 12.0 | 12.4 | 12.5 |
| EO2-NO | 34.9 | 37.2 | 37.0 |
| MEKO2-NO | 16.4 | 16.1 | 16.7 |
| 0.4*ONITR-OH | 6.0 | 6.8 | 7.1 |
| jonitr | 1.1 | 1.2 | 1.3 |
| O3-Loss | 4118.0 | 4128.9 | 4157.6 |
| O1D-H2O | 2217.8 | 2295.8 | 2290.2 |
| OH-O3 | 582.2 | 537.6 | 536.7 |
| HO2-O3 | 1203.5 | 1179.0 | 1202.4 |
| C3H6-O3 | 11.9 | 11.0 | 12.0 |
| 0.9*ISOP-O3 | 51.3 | 51.9 | 59.4 |
| C2H4-O3 | 7.9 | 8.0 | 8.1 |
| 0.8*MVK-O3 | 12.9 | 13.5 | 14.8 |
| 0.8*MACR-O3 | 2.4 | 2.4 | 2.7 |
| C10H16-O3 | 28.2 | 29.6 | 31.4 |