# Peer review of "Representation of the Community Earth System Model (CESM1) CAM4-chem within the Chemistry-Climate Model Initiative (CCMI)"

_Geoscientific Model Development, 2015_

## Referee Comment (RC1) · Anonymous Referee #1 · 8 Feb 2016

Manuscript Number: GMD-2015-237 Title: Representation of the Community Earth System Model (CESM1) CAM4-chem within the Chemistry-Climate Model Initiative (CCMI) Authors: Tilmes et al.

General Comments: This paper provides a nice overview of the CAM4-chem simulations that have been performed for CCMI. It describes the model configurations used, simulations conducted, and updates made to the model. Preliminary analyses of the model results relative to observations are also shown.

Detailed documentation of model simulations that will likely be used in a wide range of analyses through the CCMI effort is extremely useful. Someone wishing to use the model output from these simulations, but is otherwise unfamiliar with the details

concerning this model, will find this write-up to be a great reference when trying to understand how the CAM4-chem model differs from the other models participating in CCMI. All information included in this manuscript is relevant and complete for understanding the details of this model simulation, and the preliminary analysis of the results compared to observations is instructive. I therefore recommend the publication of this manuscript with minor revisions.

Specific Comments: The motivations for some of the specific changes to the model are unclear; a brief statement of why deviations from the previous version of the model or from a method described in the literature would be helpful. Instances where I'd like to see a bit more explanation include:

Page 2, Line 28 (Section 2.1): A brief mention of what issue is addressed by the improvements to the deep convection scheme (Richter and Rasch, 2008; Neale et al., 2008) would be instructive to a reader who is not so familiar with dynamics.

Page 5, Line 7 (Section 2.1.6): Is there a reason for using Leaf Area Index "from the previous model timestep instead of the average of the previous 10 days"? Is there any significant difference between biogenic emissions calculated this way versus calculated by the method of Guenther et al. (2012)?

Other Specific Comments: Page 5, Line 14 (Section 2.2): The synthetic tracers that are recommended by CCMI and included in these simulations are listed, then the O3S tracer is described. I understand that the reader could refer to the SPARC newsletter for a description of the remaining tracers, but it would be instructive to have those descriptions in this paper as well. They do not need to be defined individually, necessarily; a categorization or brief description of the usefulness of the tracers is sufficient.

Page 7, Line 25 (Section 3.1): "Differences in clouds and land surface temperatures" cause the differing VOC emissions between simulations. Prior to this, it is pointed out that REFC1SD had higher land temperatures; shouldn't higher temperatures generally lead to greater emissions of biogenic VOCs though? Does this mean clouds are

causing an even larger difference in emissions rates, if the effect of temperature is compensating? An explicit statement of why you think VOC emission rates in the SD run are so much lower than in the FR runs would be beneficial here.

Page 8, Line 16 (Section 3.2): A link between increasing methane emissions and increases in tropospheric OH is suggested here. However, the general view is that OH should decrease with increasing burdens of methane, since methane is a sink for OH. Perhaps this could be clarified.

Page 8, Line 23 (Section 3.2): "larger ozone mixing ratios in the upper troposphere in the REFC1SD experiment results in a higher oxidation capacity", however, primary production of OH in the upper troposphere is often limited by concentrations of water vapor, and so the UT has little influence on the oxidative capacity of the troposphere. Is there clear evidence in support of this conclusion? It would be helpful to state or show what led to this statement.

Page 10, Line 12 (Section 4.1.1): Why is the model overestimating winter ozone mixing ratios in the UT? STE?

Figure 1: Labels that define the colors, as in Figure 2, would be helpful here.

Technical Corrections: Page 1, Line 7 (Abstract): "observed period" is unclear; perhaps "satellite era" instead?

Page 1, Line 13 (Abstract): "has been" should be "is"

Page 2, Line 31 (Section 2.1): semi-colon between references should be an "and"

Page 3, Line 9 (Section 2.1.1): Meaning of "above 100 hPa" could be confused; suggest "at pressures less than 100 hPa" or something similar to make it absolutely clear

Page 4, Line 23 (Section 2.1.4): "black carbon and primary organic carbon, nitrates are..." should be "black carbon, primary organic carbon, and nitrates are..."

Page 5, Line 2 (Section 2.1.6): acronym used is "CLM" but was introduced as "CLM4.0"

Page 5, Line 30 (Section 2.2): Second "C" in "CAM4-Chem" should be lower case for consistency

Page 6, Line 3 (Section 2.2): semi-colon between references should be "and"

Page 6, Line 10 (Section 2.2): The "1" in "O1D" should be superscripted

Page 7, Line 5 (Section 2.3.1): Should "ran until 1959" be "ran through 1959"? The meaning conveyed is slightly different.

Page 8, Line 2 (Section 3.1): Methane lifetime due to OH reported in Supplement of Prather et al. is 11.2 years, not 11.3

Page 8, Line 7 (Section 3.1): "optical depth is with around 0.04 somewhat higher than..." should be "optical depth around 0.04 is somewhat higher than..."

Page 8, Line 15 (Section 3.2): Specify "increasing column ozone" as "increasing tropospheric column ozone"

Page 8, Line 26 (Section 3.2): Would like to see a reference here; there are plenty of candidate papers.

Page 9, Line 14 (Section 4.1.1): "altitudes below 900 hPa"—can be confusing to mix altitude and pressure coordinates; same just below in Line 16

Page 9, Line 15 (Section 4.1.1): Definition of MOZAIC acronym is not correct, compared to website

Page 10, Line 9 (Section 4.1.1): Punctuation in "U.S. . REFC1/REFC2" should be fixed

Page 10, Line 30 (Section 4.1.2): "The ozone gradient... is to the most part well captured" should be "...is for the most part well captured".

Page 11, Line 25 (Section 4.2): "the model underestimate" should be "the model underestimates"

Page 12, Line 17 (Section 4.4): "over the remote region over the Pacific" should be

"over the remote region of the Pacific"

Page 13, Line 5 (Section 5): "investiaged" should be "investigated"

Page 13, Line 18 (Section 5): remove "rather"

Figure 5, Caption: time period (1995-2011) is not consistent with time period in the text (Pg. 10, Line 14: 1995-2010)

Table A1, Title: "semi-implicit (S)" should be "semi-implicit (I)"

---

## Referee Comment (RC2) · Anonymous Referee #2 · 14 Feb 2016

Summary: This is a technical paper that summarizes the make-up and performance of the CAM4-chem model for the CCMI simulations. Publication of a paper like this is highly desirable for the CCMI models as it greatly aids the interpretation of these simulations. The paper is well-written. My major comments listed below relate to the model more than the paper; they amount to a minor revision of the paper.

At 40 km, CAM4-chem has an exceptionally low upper lid. There is some evidence in the literature that such a low lid influences stratospheric dynamics and consequently chemistry (although related factors such as differences in model physics between high- and low-top models may also influence this). By comparing CAM4-chem with the high-top version of CESM1, WACCM, it may well be possible to tease out these influences. A

comprehensive discussion of how this is reflected in the CAM4-chem behaviour would be interesting but is beyond the scope of this paper.

The authors note that there are some significant differences in model behaviour between the specified-dynamics and the free-running model. This will be of interest to an ongoing model evaluation activity which focusses on the specified-dynamics runs.

Substantial differences w.r.t. observations are found for the simulation of hydrocarbons. This could be related to the treatment of emissions, i.e. the distribution of generic "NMVOC" emissions across the primary source gases represented in the model. How is this handled here? Do you use any lumping?

Detailed comments:

P3L9: Replace "terrain-following" with "hybrid terrain-following pressure"

P3L11: This difference in vertical resolution is perhaps a little disappointing as it introduces differences into the experiments that are not directly due to the specified dynamics versus free-running experiments.

P3L12: Exactly which fields are being nudged? Do "meteorological fields" include moisture variables? How about differences in orography between the reanalysis grid and the model, which can introduce imbalances into the model? This may not be an issue if MERRA uses the same grid and orography as CAM4-chem.

P4L7ff: Does this error in the formulation of IGW mean that the model gets it right for the wrong reason? Do you have any experience with a version of the model that is not affected by this problem? The improved behaviour despite the above error suggests that either the above is true, or this process may not be important after all. Also this seems to be a new process which affects gyroscopic pumping. Do you need to change the other forms of GWD accordingly, to keep the Brewer-Dobson circulation intact?

P5L16ff: I suspect this is a misinterpretation of the formulation used by Eyring et al., SPARC Newsletter (2013). $O_3S$ is defined as identical to $O_3$ in the stratosphere but

only subject to loss but not production in the troposphere. That loss must include dry deposition otherwise the straightforward interpretation of $O_3S$ as constituting the stratospheric contribution to $O_3$ is no longer possible. The word "ozone *chemical* loss rate" used by Eyring et al. (2013) is unfortunate in this regard. Other CCMI modellers will have interpreted this differently. Also aside from the dry deposition issue, what constitutes the correct "chemical loss rate" to apply in this context is subject to an on-going debate. Which rate do you apply?

P6L20: Is that "HadISST2"? Please specify the version.

P8L7: "At 0.04, the dust optical depth is somewhat larger than. . ."

P8L14ff: This sentence is too convoluted to understand. Please rephrase / clarify.

P10L14f: Are you sure that "all model experiments reproduce observed tropospheric ozone within 20%"? This is a very far-reaching statement. I'd phrase this more carefully.

P11L25: "underestimates"

P12L5: "by up to 5 times in spring": I suggest to replace this phrase by "The model underestimates ethane by up to 80%."

P13L13: "the mid-latitude UTLS"

P13L17: replace "ascribed" with "attributed"

P13L19: replace "great" with "large an"

P16L24: "McFarlane"

---

## Referee Comment (RC3) · Anonymous Referee #3 · 22 Feb 2016

Review of: "Representation of the Community Earth System Model (CESM1) CAM4-chem within the Chemistry-Climate Model Initiative (CCMI)" by Tilmes et al.

This paper documents the configuration of CAM4-chem used in the CCMI simulations. It documents updates to CAM4-chem and compares CAM4-chem simulations to measurements in three simulation configurations. It is particularly nice that the paper documents some of the successes of CAM4-chem as well as aspects of the simulations that do not agree with measurements. In and of itself the paper offers model refinements, but does not seem to offer any particularly new model developments or new science not documented elsewhere. The interest of this paper is that it acts as a background for further analysis of the CCMI model runs and thus will be useful to the community at

large in subsequent analysis. It will be particularly useful if other modeling groups post similar papers (hopefully using similar diagnostics). I would recommend publication following minor revisions.

A few general aspects of this paper could be improved (see specific comments below). (i) Some more detail concerning differences in the model simulations should be included. (ii) In a few places the results would benefit from additional analysis. (iii) Some aspects of the paper organization detailing the simulations and model could be improved. (iv) A number of figures are put into the appendix. It is not really obvious why this is done. It just makes it harder for the reader to refer to these figures. The figures in the appendix seem as relevant as those in the main body of the paper. I would suggest including them in the main body of the text.

Comments:

1. It would be useful right in the first paragraph to specify the simulation periods for each of the CCMI simulations (REFC1, REFC1SD and REFC2).

2. P2,L12: "reference CCMI model experiments". It would be worthwhile to emphasize that this is using CAM4-chem in particular – the summarization is not for CCMI models in general.

3. The introduction does not explicitly mention model-measurement evaluation. It seems that it is important to explicitly mention this as a focus of the paper. It may be worthwhile to point out from the beginning that this paper forms the basis for a more in-depth analysis.

4. P2,L22, "The land model". It would be worthwhile stating that the version of the land model used here does not include interactive carbon and nitrogen cycling. 5. P2,L28 and P3,L25: "McFarlanle"

6. Section 2.1. Please state explicitly whether aerosols impact model photolysis.

7. Section 2.1.1. The vertical grid is in hybrid coordinates, transitioning from pure

sigma near the surface to pressure in the stratosphere.

8. P3, L13. Please state which fields are nudged and the time resolution of the input meteorological fields.

9. The QBO (2.1.2). It would be interesting to know if nudging to the QBO impacts the tropospheric chemistry simulation.

10. P5,L8. "simulated atmospheric value". It is unclear to me to what extent atmospheric CO2 is simulated: is it simulated or specified? Also, does ozone feedback onto the atmospheric radiation budget?

11. P5, L14. It doesn't make sense to me to list all these tracers in the text. Most readers will have no idea what they are. Listing in the table should suffice.

12. P6, L3. The description of lightning NOx does not really belong in the section characterizing the chemical mechanism.

13. Section 2.2 . There really is hardly any aerosol description in this section (aerosols are described in 2.1.4 and 2.1.5). Some reorganization here would make sense.

14. Section 2.3. Personally, I would put this section above to give the reader some idea of the simulations before going into details about the model. Much of the information on nudging here seems a repeat (but in more detail) of information above. Please include years of the simulations here (they are included under initial conditions and spinup below). It would be helpful if you could summarize the emission differences between the simulations and possibly put some of the 1995-2010 emission totals in Table 1. The emission differences between the simulations are important for interpreting the results. The emissions for C1SD and C1 are exactly the same, correct? This should be explicitly stated. Emissions in C1 and C1SD show much higher interannual variability than C2 (figure A1). Is this due to biomass burning emissions or something else? The reasons for this should be stated explicitly. Are there mean emission differences between C2 and C1: if so please state what these are. Are the differences between REFC1

and REFC2 over the historical period only due to differences in the emissions, or can differences be attributed to something else in addition? In summary some additional clarity in the differences between these simulations would be valuable.

15. P7, L24, Can you quote measurement estimates of SWCF? Is the REFC1SD outside the measured range?

16. P7, Figure 1. It would be helpful to know the extent that the emissions given in Figure 1 are internally calculated. Section 2.1.6 does not specify which biogenic emissions are calculated. To what extent are the emission differences in the VOCs due to those in the biogenic emissions? Do differences in biogenic emissions account for all the differences between the C1SD and C1 VOC emissions? To what extent do the biogenic emissions account for the differences between C1 and C2?

17. P8, L1, "performance". Please rephrase. I think you mean performance of the simulation, not the chemical variables.

18. P8, L13 "N", Do you mean reactive nitrogen (Nr) including NOx, PAN, N2O etc or . . .?

19. P8, L20-22, "Variations in emissions. . ..". This is a very general statement and could be elaborated. In addition to additional information on differences in emissions between the simulations to what extent can it be expected that the dynamics differ? I would assume dynamics between C1 and C2 would be similar over the historical period except for some differences in aerosol forcing. Is this correct? I am not sure what dynamical metric would be most appropriate to show? I would guess convective mass flux might be sensitive to model dynamics.

20. P8, The differences in ozone are dramatic. The high ozone values (and high OH) are notable in the SD simulation and evidently impact the methane lifetime. Instead of a general statement the authors could dig a bit deeper here – the stratospheric ozone column and lightning NOx do not seem to explain this difference in ozone. Is O3S the

same between the simulations? How about height of convection, or Hadley circulation? It would be helpful if the authors could explain the difference in these simulations more specifically.

21. Some more information on the number of ozonesondes that go into the comparision in Figure 3 would be helpful. What does the caption mean by: “12 observed profiles per year and season”? Is it 12 observed profiles per year or per season?

22. P9, L25, “Large part to differences. . ..”. Really? From Table 1 it the STE of O3 is larger in the SD simulation than the online simulations despite the fact that tropospheric O3S is smaller. Thus, the explanation given here doesn’t seem to be correct. Have the authors looked at differences in O3 loss or production between the simulations?

23. P10, It is curious that the SD simulation tends to overestimate 250 hPa ozone in the mid and high latitudes but to get about the same STE as the other simulations and to have less O3S in the troposphere. Any explanation?

24. Why is Figure A2 not used in the paper itself? It seems this figure could just as easily be included in the main paper.

25. P10, L23. What is the evidence of a transport problem (see comment 22)

26. P10, Figure 6. I find it noteworthy that the pole to mid-latitude ozone gradients are rather different in the two experiments, with the SD simulations showing a larger southward ozone gradient which seems to be more consistent with the measurements.

27. P10, L32-35. The simulated tropospheric and stratospheric total ozone using 150 hPa as a cut-off is compared to the ozone climatology based on OMI and MLS satellite observations. The authors should address to what extent we might expect an offset (possibly seasonally varying) due to an “apples to oranges” comparison. That is, what is the effect of using the 150 ppb ozone contour as a tropopause in the model versus assumptions made in the measurements? The tropospheric ozone column might be particularly sensitive to assumptions vis-à-vis the tropopause height.

28. P12, L21 “The model reproduces. . ..”? Really? This is not all clear from inspecting the figure (which is in a log-scale).

29. P12, L23, “The South-to-North gradient is represented well”? Please be more specific. Do you mean the hemispheric gradient? The aerosol burden is not always larger in the N.H. than the S.H. at all heights and months (e.g., September). This section could in general use a more in-depth and precise analysis of the model-measurement comparison for aerosols.

30. A recent paper (On the capabilities and limitations of GCCM simulations of summertime regional air quality: A diagnostic analysis of ozone and temperature simulations in the US using CESM CAM-Chem, Brown-Steiner, B.; Hess, P.G.; Lin, M.Y. (2015) Atmospheric Environment vol. 101 p. 134-148) seems to find some of the same discrepancies between specified dynamics and free-running simulations as found here.

---

## Author Comment (AC1) · 8 Apr 2016

*We thank all reviewers for helpful comments and suggestions. All comments are addressed below in a point-by-point response, indicated in italics.*

**Reviewer 1:**

General Comments: This paper provides a nice overview of the CAM4-chem simulations that have been performed for CCMI. It describes the model configurations used, simulations conducted, and updates made to the model. Preliminary analyses of the model results relative to observations are also shown.

Detailed documentation of model simulations that will likely be used in a wide range of analyses through the CCMI effort is extremely useful. Someone wishing to use the model output from these simulations, but is otherwise unfamiliar with the details concerning this model, will find this write-up to be a great reference when trying to understand how the CAM4-chem model differs from the other models participating in CCMI. All information included in this manuscript is relevant and complete for under- standing the details of this model simulation, and the preliminary analysis of the results compared to observations is instructive. I therefore recommend the publication of this manuscript with minor revisions.

Specific Comments: The motivations for some of the specific changes to the model are unclear; a brief statement of why deviations from the previous version of the model or from a method described in the literature would be helpful. Instances where I'd like to see a bit more explanation include:

Page 2, Line 28 (Section 2.1): A brief mention of what issue is addressed by the improvements to the deep convection scheme (Richter and Rasch, 2008; Neale et al., 2008) would be instructive to a reader who is not so familiar with dynamics.

*We agree with the reviewer and give some more detailed information in the text:* **"In summary, deep convection is treated by Zhang et al. (1995) with improvements in the convective momentum transport (Richter et al., 2008), which improved surface winds, stresses, and tropical convection. At the same time, an entraining plume was added to the convection parameterization, which together with the momentum transport improved the representation of the El Nino–Southern Oscillation (ENSO) significantly (Neale et al., 2008)."**

Page 5, Line 7 (Section 2.1.6): Is there a reason for using Leaf Area Index "from the previous model timestep instead of the average of the previous 10 days"? Is there any significant difference between biogenic emissions calculated this way versus calculated by the method of Guenther et al. (2012)?

*Guenther et al (2012) used monthly mean LAI maps, so in that case, the 'previous timestep' meant the average of the previous month. In the original implementation of MEGANv2.1 in CLM, this was erroneously interpreted as the previous model timestep (30 min). To be consistent with other formulas in the MEGAN algorithm (and in consultation with Alex Guenther), we corrected the CLM implementation to use LAI averaged over the previous 10 days. A corrected implementation is closer to the algorithm of Guenther et al. (2012). We changed the text to:*

**"An erroneous implementation of MEGAN in this version differs from the description of Guenther et al. (2012) by using the LAI from the previous model timestep (30 minutes) instead of the average of the previous 10 days. In addition, in this version we are using a fixed $CO_2$ mixing ratio, instead of the simulated atmospheric value, in the calculation of the $CO_2$ inhibition effect on isoprene emissions. The corrected implementation is closer to the algorithm of Guenther et al. (2012)."**

Other Specific Comments: Page 5, Line 14 (Section 2.2): The synthetic tracers that are recommended by CCMI and included in these simulations are listed, then the O3S tracer is described. I understand that the reader could refer to the SPARC newsletter for a description of the remaining tracers, but it would be instructive to have those descriptions in this paper as well. They do not need to be defined individually, necessarily; a categorization or brief description of the usefulness of the tracers is sufficient.

*We mention O3S here, because it may have been treated differently than in other models due to the interpretation of the recommendation in Eyring et al. (2013) for this tracer. We have interpreted the recommendation by CCMI to not include dry deposition for this tracer, which has to be pointed out in the text. For the other tracers, a good description is indeed given in Eyring et al., 2013. To make it easier for the reader, we give the Section number in Eyring et al., 2013, so one has an easier time finding the description.*

Page 7, Line 25 (Section 3.1): "Differences in clouds and land surface temperatures" cause the differing VOC emissions between simulations. Prior to this, it is pointed out that REFC1SD had higher land temperatures; shouldn't higher temperatures generally lead to greater emissions of biogenic VOCs though? Does this mean clouds are causing an even larger difference in emissions rates, if the effect of temperature is compensating? An explicit statement of why you think VOC emission rates in the SD run are so much lower than in the FR runs would be beneficial here.

*Without performing additional sensitivity study we can only speculate on the reason for differences between the two simulations. However, there are indications that differences in clouds play an important role. There are more low clouds in the SD simulation, as the short-*

*wave cloud forcing is a bit higher (about 30%) than the FR simulations and j-values are reduced near the surface.*

*"Differences in clouds and land surface temperatures between the reference experiments result in different biogenic emissions of volatile organic components (VOCs) (Figure~\ref{fi emis_bio}). REFC1SD biogenic emissions are about 10\% lower than in the REFC1 experiment and about 15\% lower than in the REFC2 experiment. The emissions differ the most in summer during their peak (Figure 1, bottom row). **Despite the fact that surface temperatures in REFC1SD are warmer than in REFC1, more low cloud clouds and reduced solar radiation (as evident in photolysis rates) near the surface may be the important driver for the reduced biogenic emissions in REFC1SD, which has to be further investigated.**"*

Page 8, Line 16 (Section 3.2): A link between increasing methane emissions and increases in tropospheric OH is suggested here. However, the general view is that OH should decrease with increasing burdens of methane, since methane is a sink for OH. Perhaps this could be clarified.

*This was a typo in the manuscript, methane emissions are decreasing between 1980 and 2010 and this corresponds to the increase in tropospheric OH. We fixed the text accordingly.*

Page 8, Line 23 (Section 3.2): "larger ozone mixing ratios in the upper troposphere in the REFC1SD experiment results in a higher oxidation capacity", however, primary production of OH in the upper troposphere is often limited by concentrations of water vapor, and so the UT has little influence on the oxidative capacity of the troposphere. Is there clear evidence in support of this conclusion? It would be helpful to state or show what led to this statement.

*We agree that the statement was not sufficiently supported. Other reasons for differences in methane lifetime could be changes in photolysis due to changes in high clouds. We have changed the sentence:*

*"For instance, larger ozone mixing ratios in the upper troposphere in the REFC1SD experiment results in a higher oxidation capacity of the troposphere and therefore a shorter lifetime of methane compared to the other experiments."*

*To*

*"**The shorter lifetime of methane in REFC1SD compared to the other experiments may be a result of a reduction in high clouds, and, to a small extent, larger ozone mixing ratios in the tropical troposphere, which would increase the oxidation capacity in the tropics. This has to be investigated in more detail in future studies.**"*

Page 10, Line 12 (Section 4.1.1): Why is the model overestimating winter ozone mixing ratios in

the UT? STE?

*Transport problems in the model may be the reason for the overestimation. In a follow-on study, it turns out that the nudging amount of 1% is impacting the convection in REFC1SD in a non optimal way in the troposphere (Jessica Neu, personal communication). A nudging value of 10% is improving ozone values in the UTLS. To add this information, we change the sentence to:*

*"At 250 hPa, which is in the UTLS at mid and high latitudes, REFC1SD overestimates ozone by up to 50%, particularly at mid latitudes in both hemispheres. **This could be the result of strong mixing in the UTLS associated with the use of the small nudging amount of 1% in this study; however this needs to be investigated in more detail in future studies.** The other experiments show smaller deviations from the observations of about 20% or less."*

Figure 1: Labels that define the colors, as in Figure 2, would be helpful here.

*These have been added.*

Technical Corrections: Page 1, Line 7 (Abstract): "observed period" is unclear; perhaps "satellite era" instead?

*We change this to: "We summarize the performance of the three reference simulations suggested by CCMI, **with a focus on the last 15 years of the simulation when most observations are available**."*

Page 1, Line 13 (Abstract): "has been" should be "is"

*changed*

Page 2, Line 31 (Section 2.1): semi-colon between references should be an "and"

*changed*

Page 3, Line 9 (Section 2.1.1): Meaning of "above 100 hPa" could be confused; suggest "at pressures less than 100 hPa" or something similar to make it absolutely clear

*changed*

Page 4, Line 23 (Section 2.1.4): "black carbon and primary organic carbon, nitrates are..." should be "black carbon, primary organic carbon, and nitrates are..."

*changed*

Page 5, Line 2 (Section 2.1.6): acronym used is "CLM" but was introduced as "CLM4.0"

*changed*

Page 5, Line 30 (Section 2.2): Second "C" in "CAM4-Chem" should be lower case for consistency

*changed*

Page 6, Line 3 (Section 2.2): semi-colon between references should be "and" Page 6, Line 10 (Section 2.2): The "1" in "O1D" should be superscripted

*changed*

Page 7, Line 5 (Section 2.3.1): Should "ran until 1959" be "ran through 1959"?          The meaning conveyed is slightly different.

*Changed, "through" is correct*

Page 8, Line 2 (Section 3.1): Methane lifetime due to OH reported in Supplement of Prather et al. is 11.2 years, not 11.3

*changed*

Page 8, Line 7 (Section 3.1): "optical depth is with around 0.04 somewhat higher than..." should be "optical depth around 0.04 is somewhat higher than..."

*changed*

Page 8, Line 15 (Section 3.2): Specify "increasing column ozone" as "increasing tropospheric column ozone"

*changed*

Page 8, Line 26 (Section 3.2): Would like to see a reference here; there are plenty of candidate papers.

*We added "WMO2006"*

Page 9, Line 14 (Section 4.1.1): "altitudes below 900 hPa can be confusing to mix altitude and pressure coordinates; same just below in Line 16

*This has been fixed.*

Page 9, Line 15 (Section 4.1.1): Definition of MOZAIC acronym is not correct, compared to

website

*Thanks for pointing that out, the acronym is now:* **"Measurements of OZone, water vapour, carbon monoxide and nitrogen oxides by in-service AIrbus airCraft"**

Page 10, Line 9 (Section 4.1.1): Punctuation in "U.S. . REFC1/REFC2" should be fixed

*changed*

Page 10, Line 30 (Section 4.1.2): "The ozone gradient... is to the most part well captured" should be "...is for the most part well captured".

*changed*

Page 11, Line 25 (Section 4.2): "the model underestimate" should be "the model underestimates"

*changed*

Page 12, Line 17 (Section 4.4): "over the remote region over the Pacific" should be

"over the remote region of the Pacific"

*changed*

Page 13, Line 5 (Section 5): "investiaged" should be "investigated"

*changed*

Page 13, Line 18 (Section 5): remove "rather"

*changed*

Figure 5, Caption: time period (1995-2011) is not consistent with time period in the text (Pg. 10, Line 14: 1995-2010)

*The ozonesonde climatology is derived for the period between 1995-2011, while the model results for this comparison are between 1995-2010. We clarified this in the text and the figure caption.*

**"A comparison with ozonesonde observations over different regions for simulated years between 1995-2010 …"**

Table A1, Title: "semi-implicit (S)" should be "semi-implicit (I)"

*changed*

**Reviewer 2:**

Summary: This is a technical paper that summarizes the make-up and performance of the CAM4-chem model for the CCMI simulations. Publication of a paper like this is highly desirable for the CCMI models as it greatly aids the interpretation of these simulations. The paper is well-written. My major comments listed below relate to the model more than the paper; they amount to a minor revision of the paper.

At 40 km, CAM4-chem has an exceptionally low upper lid. There is some evidence in the literature that such a low lid influences stratospheric dynamics and consequently chemistry (although related factors such as differences in model physics between high- and low-top models may also influence this). By comparing CAM4-chem with the high- top version of CESM1, WACCM, it may well be possible to tease out these influences. A comprehensive discussion of how this is reflected in the CAM4-chem behaviour would be interesting but is beyond the scope of this paper.

*We agree that this discussion is beyond the scope of this paper and a future paper will address this question. However, we have compared the CAM4-chem results with WACCM results in the troposphere, and there is very little difference. So, for tropospheric chemistry, the low-top model is behaving very similarly to the high top model.*

The authors note that there are some significant differences in model behaviour between the specified-dynamics and the free-running model. This will be of interest to an ongoing model evaluation activity which focusses on the specified-dynamics runs.

*We agree with the reviewer.  Such an activity has started after the October 2015 CCMI Workshop and is led by Clara Orbe (NASA)*

Substantial differences w.r.t. observations are found for the simulation of hydrocarbons. This could be related to the treatment of emissions, i.e. the distribution of generic "NMVOC" emissions across the primary source gases represented in the model. How is this handled here? Do you use any lumping?

*It is certainly possible that errors are introduced in the speciation of total NMVOC to the individual model species.  Emissions were provided for CCMI in a standard VOC speciation (described in Section 2.3).  In the comparisons we have made to observations, hydrocarbons are all generally under-estimated, indicating the overall emissions are too low, and that it is not purely a problem with the speciation.  The chemical species included in CAM4-chem are listed in Table A1.  The hydrocarbons ethane, ethene, ethyne, propane, propene, benzene, toluene and xylenes are treated explicitly, while BIGALK and BIGENE represent lumped alkanes and alkenes,*

*respectively, for C>3. Several VOCs are treated explicitly (CH2O, CH3CHO, CH3COCH3, CH3OH, C2H5OH), but some are lumped (e.g., MEK).*

Detailed comments:

P3L9: Replace "terrain-following" with "hybrid terrain-following pressure"

*changed*

P3L11: This difference in vertical resolution is perhaps a little disappointing as it introduces differences into the experiments that are not directly due to the specified dynamics versus free-running experiments.

*We agree that introducing a different vertical resolution without any nudging of met fields would likely change the performance of the model a bit. Past experience with changing the vertical resolution of the analysis data (to match the free-running grid) showed very significant deterioration in the quality of the simulation.*

P3L12: Exactly which fields are being nudged? Do "meteorological fields" include moisture variables? How about differences in orography between the reanalysis grid and the model, which can introduce imbalances into the model? This may not be an issue if MERRA uses the same grid and orography as CAM4-chem.

*For the SD configuration, internally derived meteorological fields, including wind components, temperature, surface pressure, surface stress, and latent and sensible heat flux are nudged to MERRA. The MERRA reanalysis fields are interpolated to the horizontal resolution of the model prior to running the simulation. The MERRA surface geopotential height is used for the SD simulations to be consistent with the reanalysis fields.*

*We adjust the text accordingly:*

*"**Nudged meteorological fields include zonal wind components, temperatures, surface pressure, surface stress, latent, and sensible heat flux. Analyzed fields are interpolated to the horizontal resolution of the model. The MERRA surface geopotential height is used for the SD simulations to be consistent with the reanalysis fields.**"*

P4L7ff: Does this error in the formulation of IGW mean that the model gets it right for the wrong reason? Do you have any experience with a version of the model that is not affected by this problem? The improved behaviour despite the above error suggests that either the above

is true, or this process may not be important after all. Also this seems to be a new process which affects gyroscopic pumping. Do you need to change the other forms of GWD accordingly, to keep the Brewer-Dobson circulation intact?

*The comparison of our simulations with those of Garcia et al (2016) and many other simulations (for testing) shows that it is important to have gravity wave drag from waves that have both relatively high momentum flux magnitudes and low horizontal phase speeds; beyond this, the details of how these waves are specified do not have much impact on the simulations as stated in the text. What seems to matter most is that the amplitude and timing of the gravity wave drag.*

P5L16ff: I suspect this is a misinterpretation of the formulation used by Eyring et al., SPARC Newsletter (2013). $O_3S$ is defined as identical to $O_3$ in the stratosphere but only subject to loss but not production in the troposphere. That loss must include dry deposition otherwise the straightforward interpretation of $O_3S$ as constituting the stratospheric contribution to $O_3$ is no longer possible. The word "ozone *chemical* loss rate" used by Eyring et al. (2013) is unfortunate in this regard. Other CCMI modellers will have interpreted this differently. Also aside from the dry deposition issue, what constitutes the correct "chemical loss rate" to apply in this context is subject to an on-going debate. Which rate do you apply?

*We agree that the wording was confusing and could also be interpreted differently. To address the comment, we change "Following the CCMI recommendation, " to "**As interpreted from the CCMI recommendation**". Regarding the chemical loss rate, we apply the definition listed on Page 6, Line 9.*

P6L20: Is that "HadISST2"? Please specify the version.  P8L7: "At 0.04, the dust optical depth is somewhat larger than. . ."  P8L14ff: This sentence is too convoluted to understand. Please rephrase / clarify.

*Changed*

P10L14f: Are you sure that "all model experiments reproduce observed tropospheric ozone within 20%"? This is a very far-reaching statement. I'd phrase this more carefully.

*We change the sentence to:*

*"**Besides some differences in ozone compared to observations, as discussed above, all model experiments reproduce observed tropospheric ozone within 25% for most of the regions**."*

P11L25: "underestimates"

*changed*

P12L5: "by up to 5 times in spring": I suggest to replace this phrase by "The model underestimates ethane by up to 80%."

*changed*

P13L13: "the mid-latitude UTLS"  P13L17: replace "ascribed" with "attributed" P13L19: replace "great" with "large an" P16L24: "McFarlane"

*changed*

**Reviewer 3:**

Review of: "Representation of the Community Earth System Model (CESM1) CAM4- chem within the Chemistry-Climate Model Initiative (CCMI)" by Tilmes et al.

This paper documents the configuration of CAM4-chem used in the CCMI simulations. It documents updates to CAM4-chem and compares CAM4-chem simulations to measurements in three simulation configurations. It is particularly nice that the paper documents some of the successes of CAM4-chem as well as aspects of the simulations that do not agree with measurements. In and of itself the paper offers model refinements, but does not seem to offer any particularly new model developments or new science not documented elsewhere. The interest of this paper is that it acts as a background for further analysis of the CCMI model runs and thus will be useful to the community at large in subsequent analysis. It will be particularly useful if other modeling groups post similar papers (hopefully using similar diagnostics). I would recommend publication following minor revisions.

A few general aspects of this paper could be improved (see specific comments below). (i) Some more detail concerning differences in the model simulations should be included. (ii) In a few places the results would benefit from additional analysis. (iii) Some aspects of the paper organization detailing the simulations and model could be improved. (iv) A number of figures are put into the appendix. It is not really obvious why this is done. It just makes it harder for the reader to refer to these figures. The figures in the appendix seem as relevant as those in the main body of the paper. I would suggest including them in the main body of the text.

   i)      *We agree with the reviewer to include more detailed information regarding the model simulations, see comments below.*
   ii)     *The scope of the paper is to document the specific model configurations for CCMI and new developments of the model. We highlight some agreements and*

> *disagreements of the model with observations. Additional analyses will be performed in future studies and multi-model analysis.*

iii) *Regarding figures in the Appendix, we tried to make the paper more concise in not including all the figures in the main paper that do not contribute to new findings. However, we would like to add them for the reader as backup information. We address the specific figures mentioned by the reviewer below.*

Comments:

1. It would be useful right in the first paragraph to specify the simulation periods for each of the CCMI simulations (REFC1, REFC1SD and REFC2).

*This information has been added.*

2. P2,L12: "reference CCMI model experiments". It would be worthwhile to emphasize that this is using CAM4-chem in particular – the summarization is not for CCMI models in *general*.

*We added CESM1 CAM4-chem*

3. The introduction does not explicitly mention model-measurement evaluation. It seems that it is important to explicitly mention this as a focus of the paper. It may be worthwhile to point out from the beginning that this paper forms the basis for a more in-depth analysis.

*The text already mentions the model-measurement evaluation:*

*".. and evaluate selected diagnostics based on observational data sets in Section 4. We employ existing and new datasets to evaluate the general performance of the model."*

*To address the comment, we add:*

*"More in-depth analysis and evaluations will follow in multi-model comparison studies."*

4. P2,L22, "The land model". It would be worthwhile stating that the version of the land model used here does not include interactive carbon and nitrogen cycling.

*The version of the land model can be run with interactive carbon and nitrogen cycle, however, in our configuration, this was not included.*

*We change "The land model does not include an interactive carbon or nitrogen cycle and only the atmospheric and land components are coupled to the chemistry."*

*To*

*"The land model was **run without** an interactive carbon or nitrogen cycle and only the atmospheric and land components are coupled to the chemistry."*

5. P2,L28 and P3,L25: "McFarlanle"

*corrected*

6. Section 2.1. Please state explicitly whether aerosols impact model photolysis.

*We added the following:*

*"**Only changes in the ozone column, but not in the aerosol burden, impact photolysis rates.**"*

7. Section 2.1.1. The vertical grid is in hybrid coordinates, transitioning from pure sigma near the surface to pressure in the stratosphere.

*We changed the text:*

*"**The vertical coordinate is sigma hybrid terrain-following pressure in the troposphere, switching over to isobaric at pressure levels less than 100 hPa;**"*

8. P3, L13. Please state which fields are nudged and the time resolution of the input meteorological fields.

*We added to the text:*

*"**Nudged meteorological fields include wind components, temperatures, surface pressure, surface stress, latent, and sensible heat flux. Analyzed fields are interpolated to the horizontal resolution of the model. The MERRA surface geopotential height is used for the SD simulations to be consistent with the reanalysis fields.**"*

9. The QBO (2.1.2). It would be interesting to know if nudging to the QBO impacts the tropospheric chemistry simulation.

*This is an interesting question, which however cannot be addressed in this paper.*

10. P5,L8. "simulated atmospheric value". It is unclear to me to what extent atmospheric $CO_2$ is simulated: is it simulated or specified? Also, does ozone feedback onto the atmospheric radiation budget?

*We add a clarification on emissions, lower boundary conditions, and radiatively active species in this section:*

*"Emissions of gas-phase and aerosol species, as indicated in Table A1, are in general distributed at the surface. Only aircraft emissions of BC and nitrogen dioxide, and volcanic emissions of sulfur and sulfate, are vertically distributed. Species with lower boundary conditions, listed in Table A1, as discussed in Section 2.3.2."*

*and later on*

*"All aerosols and some gas-phase species, including H2O, O2, CO2, O3, N2O, CH4, CFC11, CFC12, are radiatively active."*

11. P5, L14. It doesn't make sense to me to list all these tracers in the text. Most readers will have no idea what they are. Listing in the table should suffice.

*We list the tracers to indicate which ones have been included. We also add a detailed reference, Eyring et al., 2013, Section 4.2.1, to make it easy for the reader to look up the meaning of the tracers.*

12. P6, L3. The description of lightning NOx does not really belong in the section characterizing the chemical mechanism.

*We agree and move it to the atmosphere model section.*

13. Section 2.2 . There really is hardly any aerosol description in this section (aerosols are described in 2.1.4 and 2.1.5). Some reorganization here would make sense.

*We agree with the reviewer and rename section 2.2 to: "Chemical mechanism" and move the one sentence on aerosols to Section 2.1.5.*

14. Section 2.3. Personally, I would put this section above to give the reader some idea of the simulations before going into details about the model. Much of the information on nudging here seems a repeat (but in more detail) of information above.

*The experimental setup of the model is independent of the model description. Other experiments, for example HTAP, would use the same model code. Therefore, we think it makes more sense to keep it separated.*

Please include years of the simulations here (they are included under initial conditions and spinup below). It would be helpful if you could summarize the emission differences between the simulations and possibly put some of the 1995-2010 emission totals in Table 1. The emission differences between the simulations are important for interpreting the results. The emissions for C1SD and C1 are exactly the same, correct? This should be explicitly stated.

Emissions in C1 and C1SD show much higher interannual variability than C2 (figure A1). Is this due to biomass burning emissions or something else? The reasons for this should be stated explicitly. Are there mean emission differences between C2 and C1: if so please state what these are.

*The emissions for the different experiments are shown in Figure A1. The time evolution and differences in variability are more obvious if shown in a figure than giving total numbers. We adjust the text to clarify differences between the emissions.*

***"The three reference experiments are performed with the recommended emissions. REFC1 and REFC1SD (years 1960-2010), use the same emissions, excluding biogenic emissions. Anthropogenic and biomass burning emissions are from the MACCity emission data set and change every year Granier et al. (2011). For REFC2 (years 1960-2100), anthropogenic and biomass burning emissions are taken from AR5 (Eyring et al., 2013) (see Figure A1), which only vary every 5-10 years. All emissions include a seasonal cycle. All biogenic emissions are calculated every timestep by MEGAN, as described in Section 2.1.6."***

Are the differences between REFC1 and REFC2 over the historical period only due to differences in the emissions, or can differences be attributed to something else in addition? In summary some additional clarity in the differences between these simulations would be valuable.

*As can be seen from Figure A1 differences in emissions will have some impact. However, since REFC2 is couple to the ocean, the climate will also be different, which has of course a large impact on the atmospheric composition.*

15. P7, L24, Can you quote measurement estimates of SWCF? Is the REFC1SD outside the measured range?

*Observed global numbers of SWCF are between 47 W/m2 (CERES) and 54 W/m2 (ERBE). We adjusted the text:*
***"In the REFC1SD experiment, low cloud fraction is significantly larger than in the other experiments, which results in a much smaller shortwave cloud forcing (SWCF) of -83 W/m2 compared the other experiments that are with 54-56 W/m2 more in line with observations."***

16. P7, Figure 1. It would be helpful to know the extent that the emissions given in Figure 1 are internally calculated. Section 2.1.6 does not specify which biogenic emissions are calculated. To what extent are the emission differences in the VOCs due to those in the biogenic emissions? Do differences in biogenic emissions account for all the differences between the C1SD and C1 VOC emissions? To what extent do the biogenic emissions account for the differences between

C1 and C2?

*All biogenic emissions are internally calculated and therefore account for all the differences in the emissions between REFC1 and REFC1SD, as clarified above. Figure 1, left top panel, shows all VOC emissions, while the second left top panel shows only the biogenic part of the VOC emissions. As can be seen from that figure, the differences between REFC1 and REFC2 in biogenic emissions are much smaller than between REFC1 and REFC1SD. We add a sentence regarding the cause of the differences in emissions at Page 7, last paragraph:*

*"The emissions differ the most in summer during their peak (Figure 1, bottom row). **Despite the fact that surface temperatures in REFC1SD are warmer than in REFC1, more low cloud clouds and reduced solar radiation (as evident in photolysis rates) near the surface may be the important driver for the reduced biogenic emissions in REFC1SD, which has to be further investigated.***

17. P8, L1, "performance". Please rephrase. I think you mean performance of the simulation, not the chemical variables.

*Changed*

18. P8, L13 "N", Do you mean reactive nitrogen (N) including NOx, PAN, N2O etc or …?

*We looked at NO2 and clarified this in the text. Figure 2 shows NO2 but is in units TgN. We changed the figure accordingly.*

19. P8, L20-22, "Variations in emissions....". This is a very general statement and could be elaborated. In addition to additional information on differences in emissions between the simulations to what extent can it be expected that the dynamics differ? I would assume dynamics between C1 and C2 would be similar over the historical period except for some differences in aerosol forcing. Is this correct? I am not sure what dynamical metric would be most appropriate to show? I would guess convective mass flux might be sensitive to model dynamics.

*To investigate differences in methane lifetime, various effects have to be taken into account and differences in dynamics and convection, but also nudging of the model, may play an important role. All this has to be addressed in a future study.*

20. P8, The differences in ozone are dramatic. The high ozone values (and high OH) are notable in the SD simulation and evidently impact the methane lifetime. Instead of a general statement the authors could dig a bit deeper here – the stratospheric ozone column and lightning NOx do

not seem to explain this difference in ozone. Is O3S the same between the simulations? How about height of convection, or Hadley circulation? It would be helpful if the authors could explain the difference in these simulations more specifically.

*These are interesting questions and will need more investigation, but they are beyond the scope of this paper. We do add at Page 10 Line 12, based on a response to Reviewer 1:*

*"At 250 hPa, which is the UTLS at mid and high latitudes, REFC1SD overestimates ozone by up to 50%, particularly at mid latitudes in both hemispheres. **This could be the result of strong mixing in the UTLS associated with the use of the small nudging amount of 1% in this study; however, this needs to be investigated in more detail in future studies.**" The other experiments show smaller deviations from the observations of about 20% or less."*

21. Some more information on the number of ozonesondes that go into the comparison in Figure 3 would be helpful. What does the caption mean by: "12 observed profiles per year and season"? Is it 12 observed profiles per year or per season?

*Changed to "**12 observed profiles per season in a year**".*

22. P9, L25, "Large part to differences. . ..". Really? From Table 1 it the STE of O3 is larger in the SD simulation than the online simulations despite the fact that tropospheric O3S is smaller. Thus, the explanation given here doesn't seem to be correct. Have the authors looked at differences in O3 loss or production between the simulations?

*We agree, that STE difference may not be the largest part of the difference. We also have looked at O3 loss and production for the same regions, and do see differences there as well, and change the sentence to:*

*"Results from REFC1 and REFC2 show larger deviations from the observations than REFC1SD over these two regions. **These are in part due to differences in the amount of stratospheric ozone entering the troposphere for the different experiments (see Figure 3, right column, dashed lines), but also due to changes in ozone loss and production, especially in summer.** Discrepancies in ozone between the experiments can be explained by differences in O3S for the whole year at 500 hPa and for winter months at 900 hPa. During summer months, differences in chemical production at the surface for the different experiments seem to play an additional role and explain about 5-10 ppb of the deviations for Western Europe."*

23. P10, It is curious that the SD simulation tends to overestimate 250 hPa ozone in the mid and high latitudes but to get about the same STE as the other simulations and to have less O3S in the troposphere. Any explanation?

*Reviewer 1 raised a similar concern, we answered and changed the text (see above):*

*Transport problems in the model may be the likely reason for the overestimation. In follow-on studies, it turns out that the nudging amount of 1% is impacting the convection in REFC1SD in a non optimal way in the troposphere (Jessica Neu, personal communication). A nudging value of 10% is improving ozone values in the UTLS.*

24. Why is Figure A2 not used in the paper itself? It seems this figure could just as easily be included in the main paper.

*We have included a lot of discussion on ozone and there is little new that this figure offers. We added it for completeness in the supplement.*

25. P10, L23. What is the evidence of a transport problem (see comment 22)

*As discussed above, larger differences in O3S do occur in high northern latitudes in winter and spring.*

26. P10, Figure 6. I find it noteworthy that the pole to mid-latitude ozone gradients are rather different in the two experiments, with the SD simulations showing a larger southward ozone gradient which seems to be more consistent with the measurements.

*We change the text to:*

**"Observed features, for example the summertime maximum of ozone over the eastern Mediterranean/Middle East (Kalabokas et al., 2013, Zanis et al., 2014), are reproduced by the REFC1 and REFC1SD experiments. The ozone gradient between mid latitudes and tropics is for the most part well captured, for example over Japan in summer. The pole to mid-latitude ozone gradient in the SD simulation is showing a larger southward ozone gradient than the REFC1 simulation, which is more consistent with the measurements.**
*Regional differences in tropospheric ozone between the different model experiments have to be investigated in future studies."*

27. P10, L32-35. The simulated tropospheric and stratospheric total ozone using 150 hPa as a cut-off is compared to the ozone climatology based on OMI and MLS satellite observations. The authors should address to what extent we might expect an offset (possibly seasonally varying) due to an "apples to oranges" comparison. That is, what is the effect of using the 150 ppb ozone contour as a tropopause in the model versus assumptions made in the measurements? The tropospheric ozone column might be particularly sensitive to assumptions vis-à-vis the tropopause height.

*There are small differences in choosing the tropopause level. However, comparisons between the simulations are based on the same criteria. We change the text to:*

**"The model tropopause for this diagnostic is defined as the 150 ppb ozone level, which may lead to small differences between observations and model simulations, but not between model experiments themselves**.*"*

28. P12, L21 "The model reproduces. . ..."? Really? This is not all clear from inspecting the figure (which is in a log-scale).

*The uploaded figure had a problem, in particular of the low altitude averages. We include the new figure in the revised manuscript, the model simulations agree somewhat better in mid- to high latitudes. We also change the sentence pointed out by the reviewer to:*

*"The model reproduces BC values in the SH and NH mid latitudes* **for most seasons within the range of uncertainty.** *"*

29. P12, L23, "The South-to-North gradient is represented well"? Please be more specific. Do you mean the hemispheric gradient? The aerosol burden is not always larger in the N.H. than the S.H. at all heights and months (e.g., September). This section could in general use a more in-depth and precise analysis of the model-measurement comparison for aerosols.

*We think, that most of the important information has been included in the paragraph, and to clarify change the paragraph to:*

*"Otherwise,* **in spring and summer***, the* **hemispheric** *gradient of BC is represented well, following the observed larger burden in the NH compared to the SH, with some overestimation in the SH. The largest BC values in the NH spring are however underestimated. On the other hand, BC values in August/September, and partly November, are overestimated in the NH and in March/April and June/July in the SH."*

30. A recent paper (On the capabilities and limitations of GCCM simulations of summertime regional air quality: A diagnostic analysis of ozone and temperature simulations in the US using CESM CAM-Chem, Brown-Steiner, B.; Hess, P.G.; Lin, M.Y. (2015) Atmospheric Environment vol. 101 p. 134-148) seems to find some of the same discrepancies between specified dynamics and free-running simulations as found here.

*Papers that have used the version CESM1.2.2 or similar updates show similar discrepancies. The paper that discusses those is Tilmes et al., 2015. We are adding this reference to the conclusions.*